# Distilling Model Failures as Directions in Latent Space

**Saachi Jain\*, Hannah Lawrence\*, Ankur Moitra & Aleksander Mądry**
Massachusetts Institute of Technology, Cambridge, USA
`{saachij,hanlaw,moitra,madry}`@mit.edu, (\*Equal contribution)

## Abstract

Existing methods for isolating hard subpopulations and spurious correlations in datasets often require human intervention. This can make these methods labor-intensive and dataset-specific. To address these shortcomings, we present a scalable method for automatically distilling a model's failure modes. Specifically, we harness linear classifiers to identify consistent error patterns, and, in turn, induce a natural representation of these failure modes as *directions within the feature space*. We demonstrate that this framework allows us to discover and automatically caption challenging subpopulations within the training dataset. Moreover, by combining our framework with off-the-shelf diffusion models, we can generate images that are especially challenging for the analyzed model, and thus can be used to perform synthetic data augmentation that helps remedy the model's failure modes.

## 1 Introduction

The composition of the training dataset has key implications for machine learning models' behavior (Feldman, 2019; Carlini et al., 2019; Koh & Liang, 2017; Ghorbani & Zou, 2019; Ilyas et al., 2022), especially as the training environments often deviate from deployment conditions (Rabanser et al., 2019; Koh et al., 2020; Hendrycks et al., 2020). For example, a model might struggle on specific subpopulations in the data if that subpopulation was mislabeled (Northcutt et al., 2021; Stock & Cisse, 2018; Beyer et al., 2020; Vasudevan et al., 2022), underrepresented (Sagawa et al., 2020; Santurkar et al., 2021), or corrupted (Hendrycks & Dietterich, 2019; Hendrycks et al., 2020). More broadly, the training dataset might contain spurious correlations, encouraging the model to depend on prediction rules that do not generalize to deployment (Xiao et al., 2020; Geirhos et al., 2020; DeGrave et al., 2021). Moreover, identifying meaningful subpopulations within data allows for dataset refinement (such as filtering or relabeling) (Yang et al., 2019; Stock & Cisse, 2018), and training more fair (Kim et al., 2019; Du et al., 2021) or accurate (Jabbour et al., 2020; Srivastava et al., 2020) models.

However, dominant approaches to such identification of biases and difficult subpopulations within datasets often require human intervention, which is typically labor intensive and thus not conducive to routine usage. For example, recent works (Tsipras et al., 2020; Vasudevan et al., 2022) need to resort to manual data exploration to identify label idiosyncrasies and failure modes in widely used datasets such as ImageNet. On the other hand, a different line of work (Sohoni et al., 2020; Nam et al., 2020; Kim et al., 2019; Liu et al., 2021; Hashimoto et al., 2018) does present automatic methods for identifying and intervening on hard examples, but these methods are not designed to capture *simple, human-understandable* patterns. For instance, Liu et al. (2021) directly upweights inputs that were misclassified early in training, but these examples do not necessarily represent a consistent failure mode. This motivates the question:

*How can we extract meaningful patterns of model errors on large datasets?*

One way to approach this question is through model interpretability methods. These include saliency maps (Adebayo et al., 2018; Simonyan et al., 2013), integrated gradients (Sundararajan et al., 2017), and LIME (Ribeiro et al., 2016b), and perform feature attribution for particular inputs. Specifically, they aim to highlight which parts of an input were most important for making a model prediction, and can thus hint at brittleness of that prediction. However, while feature attribution can indeed help debug individual test examples, it does not provide a global understanding of the underlying biases of the dataset — at least without manually examining many such individual attributions.

OUR CONTRIBUTIONS

In this work, we build a scalable mechanism for globally understanding large datasets from the perspective of the model's prediction rules. Specifically, our goal is not only to identify interpretable failure modes within the data, but also to inform actionable interventions to remedy these problems.

Our approach distills a given model's failure modes as *directions* in a certain feature space. In particular, we train linear classifiers on normalized feature embeddings within that space to identify consistent mistakes in the original model's predictions. The decision boundary of each such classifier then defines a "direction" of hard examples. By measuring each data-point's alignment with this identified direction, we can understand how relevant that example is for the failure mode we intend to capture. We leverage this framework to:

- *Detection:* Automatically detect and quantify reliability failures, such as brittleness to distribution shifts or performance degradation on hard subpopulations (Section 2.1).
- *Interpretation:* Understand and automatically assign meaningful captions to the error patterns identified by our method (Section 2.2).
- *Intervention:* Intervene during training in order to improve model reliability along the identified axes of failure (Section 2.3). In particular, by leveraging our framework in conjunction with off-the-shelf diffusion models, we can perform synthetic data augmentation tailored to improve the analyzed model's mistakes.

Using our framework, we can automatically identify and intervene on hard subpopulations in image datasets such as CIFAR-10, ImageNet, and ChestX-ray14. Importantly, we do not require direct human intervention or pre-annotated subgroups. The resulting framework is thus a scalable approach to identifying important subpopulations in large datasets with respect to their downstream tasks.

## 2 CAPTURING FAILURE MODES AS DIRECTIONS WITHIN A LATENT SPACE

The presence of certain undesirable patterns, such as spurious correlations or underrepresented subpopulations, in a training dataset can prevent a learned model from properly generalizing during deployment. As a running example, consider the task of distinguishing "old" versus "young" faces, wherein the training dataset age is spuriously correlated with gender (such that the faces of older men and younger women are overrepresented). Such correlations occur in the CelebA dataset (Liu et al., 2015) (though here we construct a dataset that strengthens them)[1]. Thus, a model trained on such a dataset might rely too heavily on gender, and will struggle to predict the age of younger men or older women. How can we detect model failures on these subpopulations?

The guiding principle of our framework is to model such failure modes as *directions* within a certain latent space (Figure 1). In the above example, we would like to identify an axis such that the (easier) examples of "old men" and the (harder) examples of "old women" lie in opposite directions. We then can capture the role of an individual data point in the dataset by evaluating how closely its normalized embedding aligns with that extracted direction (axis). But how can we learn these directions?

**Our method.** The key idea of our approach is to find a *hyperplane* that best separates the correct examples from incorrect ones. In the presence of global failure modes such as spurious correlations, the original model will likely make consistent mistakes, and these mistakes will share features. Using a held out validation set, we can therefore train a linear support vector machine (SVM) for each class to predict the original model's mistakes based on these shared features. The SVM then establishes a decision boundary between the correct and incorrect examples, and the direction of the failure mode will be orthogonal to this decision boundary (i.e., the normal vector to the hyperplane). Intuitively, the more aligned an example is with the identified failure direction, the harder the example was for the original neural network. Details of our method can be found in Appendix A.

**The choice of latent space: Leveraging shared vision/language embeddings.** Naturally, the choice of embedding space for the SVM greatly impacts the types of failure modes it picks up. Which embedding space should we choose? One option is to use the latent space of the original neural network. However, especially if the model fits the training data perfectly, it has likely learned latent

---

[1]We can also detect this failure mode in the original CelebA dataset (See Appendix B.1)

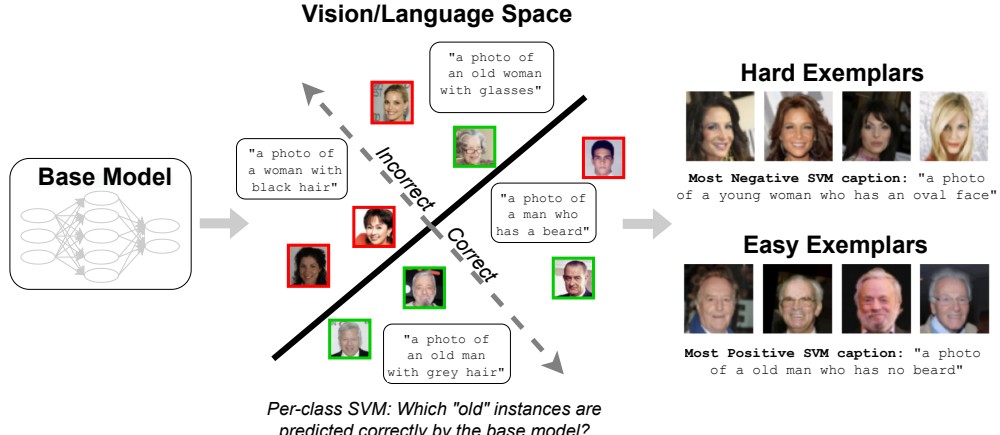

Figure 1: A summary of our approach, as applied to CelebA. Consider the task of classifying faces as "young" or "old", where the training set contains a spurious correlation with gender. (**Left**) Given a trained base model, we evaluate it on a held-out validation set. (**Middle**) For each class (here "old"), we train a linear SVM on a shared vision/language latent space to predict whether the base model would have classified an input from the validation set correctly. We then extract a *direction* (gray) for the captured failure mode as the vector orthogonal to the learned hyperplane. Here, the SVM learns to use gender to separate the incorrect (red) vs. correct (green) examples. (**Right**) Images farthest from the SVM decision boundary exemplify the hardest ("old women") or the easiest ("old men") test examples. Furthermore, we can select captions that, when embedded in the shared latent space, are farthest from the decision boundary and thus capture the pattern of errors learned by the SVM.

representations which overfit to the training labels. In our running example of CelebA, the few older women in the training set could be memorized, and so their latent embeddings would likely be very different from those of older women in the test set. As shown in Appendix B.1.6, these discontinuities reduce the efficacy of our method, as the resulting featurizations are likely inconsistent.

To address this problem, we use an embedding that is agnostic to the specific dataset. In particular, we featurize our images using CLIP (Radford et al., 2021), which embeds both images and language into a shared latent space of unit vectors. Using these embeddings will also let us automatically assign captions to the directions extracted from the SVM. We consider other latent spaces in Appendix B.1.

Having captured the model's failure modes as directions within that latent space, we can detect, interpret, and intervene on difficult subpopulations. We now describe implementing these primitives.

## 2.1 DETECTION

How can we tell whether the extracted direction actually encapsulates a prevalent error pattern? To answer this question, we need a way to quantify the strength of the identified failure mode. Our framework provides a natural metric: the validation error of the trained SVMs. The more consistent the failure mode is, the more easily a simple linear classifier can separate the errors in the CLIP embedding space. Thus, the cross-validation score (i.e, the SVM's accuracy on held-out data) serves as a measure of the failure mode's strength in the dataset. It can then be used to detect classes that have a clear bias present, which will be useful for the following interpretation and intervention steps.

## 2.2 INTERPRETATION

Since the trained SVMs are constrained to capture simple (approximately linearly separable) patterns in embedding space, we can easily interpret the extracted directions to understand the failure modes of the dataset. We explore two approaches to extracting the subpopulations captured by our framework.

**Most aligned examples.** The examples whose normalized embeddings are most aligned with the extracted direction represent the most prototypically correct or incorrect inputs. Therefore, to get a sense of what failure mode the direction is capturing, we can examine the most extreme examples according to the SVM's *decision value*, which is proportional to the signed distance to the SVM's

decision boundary. Returning to our running example, in Figure 1 the images of the "old" class that are the "most correct" correspond to men, while the "most incorrect" ones correspond to women.

**Automatic captioning.** We can leverage the fact that the SVM is trained on a shared vision/language latent space (i.e., CLIP) to automatically assign captions to the captured failure mode. Just as above we surfaced the most aligned *images*, we can similarly surface the most aligned *captions*, i.e, captions whose normalized embedding best matches the extracted direction.

Specifically, assume that each class has a reference caption $r$, which is a generic phrase that could describe all examples of the class (e.g., "a photo of a person"). We then generate a candidate set of more specific captions $c_1, ..., c_m$ that include additional attributes (e.g., "a photo of a person with a moustache"). Our goal is to pick the caption for which the *additional* information provided by $c$ — beyond that which was already provided by $r$ — best captures the the extracted failure mode.

To do so, we score a caption $c$ by how aligned the normalized embedding $\hat{c} = \frac{c-r}{||c-r||_2}$ is with the direction captured by the SVM. This amounts to choosing the captions for which $\hat{c}$ has the most positive (easiest) or negative (hardest) SVM decision values. In contrast to previous works that choose the captions closest to the mean of a selected group of hard examples (c.f., Eyuboglu et al. (2022)), our method avoids this proxy and directly assigns a caption to the captured failure mode itself.

**Directly decoding the SVM direction with diffusion models.** Some of the recently developed diffusion models (e.g., retrieval augmented diffusion models (Blattmann et al., 2022), DALL-E 2 (Ramesh et al., 2022)) generate images directly from the shared vision/language (CLIP) space. In such cases, we can *directly* decode the extracted SVM direction into generated images. This enables us to visually capture the direction itself, without needing to generate a set of candidate captions.

Specifically, let $r$ be the normalized embedding of the reference caption, and $w$ the normalized SVM direction. By rotating between $r$ and either $w$ or $-w$ via spherical interpolation, we can generate harder or easier images, respectively. Here, we expect the degree of rotation $\alpha$ to determine the extent of difficulty[2]. As shown in Section 4, passing this rotated embedding to the diffusion model indeed lets us directly generate images that encapsulate the extracted failure mode.

## 2.3 INTERVENTION

Now that we have identified some of the failure modes of the model, can we improve our model's performance on the corresponding challenging subpopulations? It turns out that this is possible, via both real and synthetic data augmentation.

**Filtering intervention.** Given an external pool of examples that was not used to train the original model, we can select the best of these examples to improve performance on the hard subpopulations. Specifically, if the goal is to add only $K$ examples per class to the training dataset (e.g., due to computation constraints), we simply add the $K$ images with the most negative SVM decision values. In Appendix B.1.2, we discuss an alternative intervention scheme, upweighting hard training examples.

**Synthetic data augmentation.** In the absence of such an external pool of such data, we can leverage our framework together with text-to-image diffusion models (e.g., Stable Diffusion (Rombach et al., 2022), DALL-E 2 (Ramesh et al., 2022), and Imagen (Saharia et al., 2022)) to generate synthetic images instead. After automatically captioning each failure mode, we simply input these captions into the diffusion model to generate images from the corresponding subpopulation. In Section 4, we show that fine-tuning the model on such images improves model reliability on these subpopulations.

## 3 VALIDATING OUR FRAMEWORK IN THE PRESENCE OF KNOWN SUBPOPULATIONS

In Section 2, we presented an approach for distilling the failure modes as directions within a latent space. In this section, we validate our framework by evaluating its performance on datasets with *known* pathologies. In Section 4, we apply our framework to discover challenging subpopulations in datasets where the bias is *not* known beforehand. Experimental details can be found in Appendix B.

---

[2]Our approach mirrors the "text-diff" technique in DALLE-2 (Ramesh et al., 2022), which uses spherical interpolation to transform images according to textual commands.

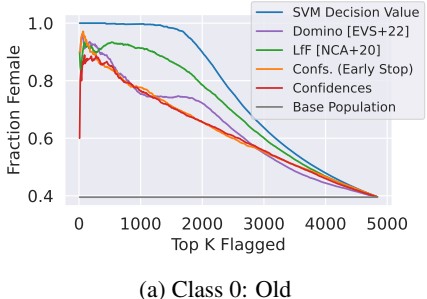
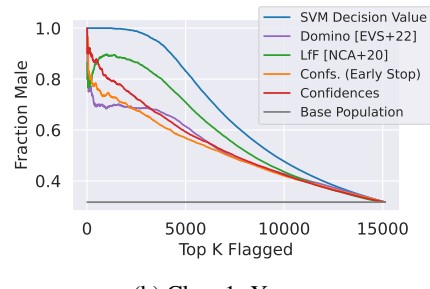

(a) Class 0: Old  (b) Class 1: Young

Figure 2: For each CelebA class, the fraction of test images that are of the minority gender when ordering the images by either their SVM decision value or model confidences. We include additional baselines: Domino (Eyuboglu et al., 2022), LfF (Nam et al., 2020), and confidences after early stopping. Our framework more reliably captures the spurious correlation than all other baselines.

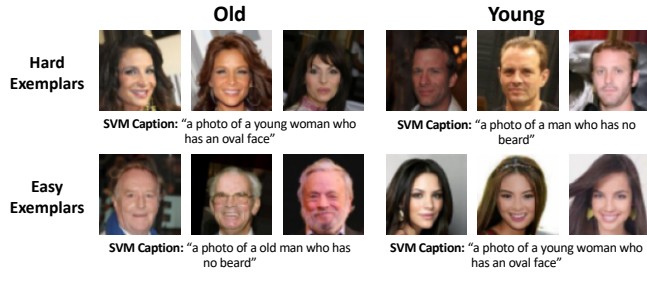

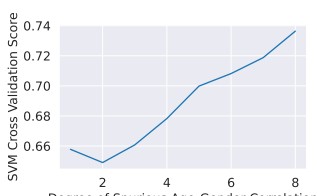

Figure 3: The images and captions for each class with the most extreme SVM decision values. Those scored as most incorrect are in the minority subpopulations ("old women" and "young men"), while those scored as most correct are in the majority subpopulations ("old men" and "young women").

Figure 4: Our SVM's cross validation score (corresponding to its estimated ability to assess the strength of the extracted failure mode) compared to the strength of the planted spurious correlation. The SVM scores are highly correlated with the strength of the shift.

We focus on two settings: (1) detecting a spurious correlation in a facial recognition dataset, and (2) isolating underrepresented subtypes in image classification. In Appendix B, we consider other settings, such as Colored MNIST (Arjovsky et al., 2019) and ImageNet-C (Hendrycks & Dietterich, 2019).

### 3.1 USING OUR FRAMEWORK TO ISOLATE AND LABEL SPURIOUS CORRELATIONS

We revisit our running example from Section 2, where a model is trained to predict age from the CelebA dataset. The training dataset contains a spurious correlation with gender (which we enhance by selectively subsetting the original dataset). We use a validation set that is balanced across age and gender, but also explore unbalanced validation sets in Appendix B.1 and Section 4.

**Capturing the spurious correlation.** Does our framework identify gender as the key failure mode in this setting? It turns out that even though only 33% of the dataset comes from the challenging subpopulations "old women" and "young men", 89% of the images flagged as incorrect by our framework are from these groups. Indeed, as shown in Figure 2, the SVM more consistently selects those hard subpopulations, compared to using the original confidences. As the underlying linear classifier is forced to use simple prediction rules, the corresponding SVM flags a more homogenous population than does using the original confidences. We find that our method further outperforms other baselines such as Domino (Eyuboglu et al., 2022), Learning from Failure (Nam et al., 2020), or using confidences after early stopping (as in Liu et al. (2021)) (see Appendix B.1 for further experimental details).

**Automatically interpreting the captured failure mode.** In Figure 3, we surface the examples most aligned with the extracted direction using the SVM's decision values. As shown, examples with the most negative SVM decision values are indeed from the hard subpopulations. We also automatically caption each captured failure mode. To this end, we consider a candidate caption set which includes attributes such as age, gender, and facial descriptors. The captions that are most aligned with the extracted direction in CLIP space capture the spurious correlation on gender (Figure 3).

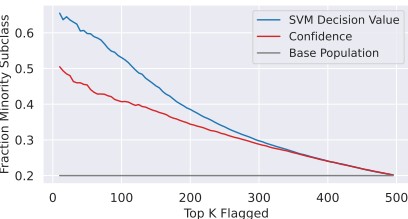
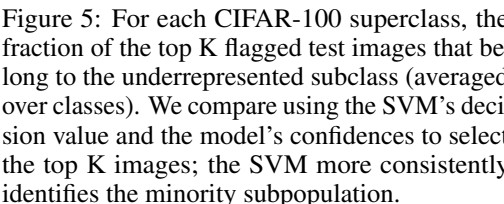
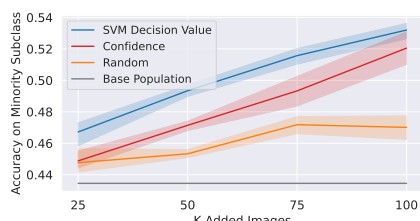

Figure 5: For each CIFAR-100 superclass, the fraction of the top K flagged test images that belong to the underrepresented subclass (averaged over classes). We compare using the SVM's decision value and the model's confidences to select the top K images; the SVM more consistently identifies the minority subpopulation.

Figure 6: For CIFAR-100, the accuracy on the minority subclasses after adding K images per superclass from the extra data (reported over five runs.) We compare using the SVM's decision value or the model's confidences to select images. Relying on the SVM's values provides the most improvement on the minority subclasses.

**Quantifying the strength of the shift.** As mentioned in Section 2, the cross-validation scores of the SVM quantify the strength of the spurious correlation. In Figure 4, we train base models on versions of CelebA with increasing degrees of the planted spurious correlation, and find that the SVM cross-validation scores strongly correlate with the strength of the planted shift.

## 3.2 IDENTIFYING UNDERREPRESENTED SUB-TYPES

Above, we considered a setting where the model predicts incorrectly because it relies on a spurious feature (gender). What if a model struggles on a subpopulation not due to a spurious feature, but simply because there are not enough examples of that image type?

To evaluate our approach in this setting, consider the task of predicting the superclass of a hierarchical dataset when subclasses have been underrepresented. In particular, the CIFAR-100 (Krizhevsky, 2009) dataset contains twenty superclasses, each of which contains five equally represented subclasses. For each such superclass, we subsample one subclass so that it is underrepresented in the training data. For instance, for the superclass "aquatic mammals", we remove a large fraction of the subclass beavers from the training dataset. In Figure 5, we find that our framework correctly isolates the minority subclass as the most incorrect for each superclass more consistently than does confidences.

**Validating automatic captioning.** For each superclass, we construct a caption set that corresponds to the possible subclasses (e.g., for the superclass "aquatic mammals," the caption set is "a photo of a beaver", "a photo of a dolphin", etc.). As we find, the caption corresponding to the minority subclass was the top negative caption for 80% of the superclasses, and in the top 2 for 95% of the superclasses.

**Filtering intervention.** Having isolated hard subpopulations, we now apply our framework downstream to improve performance on them. We use the trained SVMs to choose a subset of images from a larger pool of data to add to the training set. In Figure 6, we find that using our framework to select this extra data improves accuracy on the minority subclasses more than what using model confidences or choosing a random subset offers, while maintaining approximately the same overall accuracy.

## 4 DISCOVERING NEW SUBPOPULATIONS IN IMAGE DATASETS

Above, we considered datasets with pathologies, such as spurious correlations or underrepresented subtypes, that were known. In this section, we apply our framework to datasets without any pre-annotated difficult subpopulations. Our approach discovers new subpopulations representing key failure modes for the models. Specifically, we apply our framework to CIFAR-10 (Krizhevsky, 2009) and ImageNet (Deng et al., 2009), and defer ChestX-ray14 (Rajpurkar et al., 2017) to Appendix C.3. Experimental details can be found in Appendix C.

### 4.1 CIFAR-10

We begin with the CIFAR-10 (Krizhevsky, 2009) dataset. Since the accuracy of a ResNet18 on CIFAR-10 is very high (93%), the original model does not make many errors. Thus, we instead

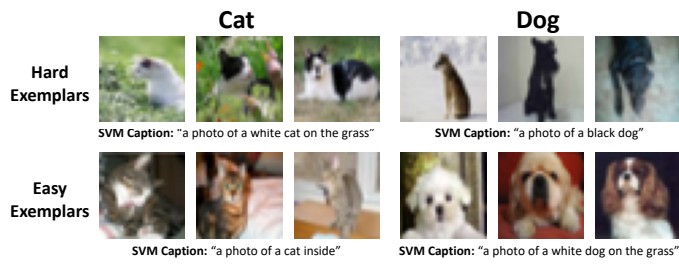

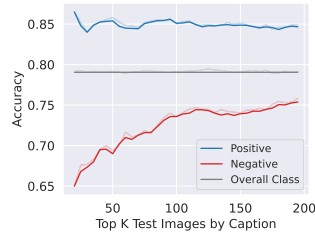

(a) Most extreme images and extracted SVM captions

(b) Model accuracies on each identified subpopulation.

Figure 7: (**a**) The images with the most extreme SVM decision values for the CIFAR-10 classes cat and dog, along with the most positive/negative captions according to the SVM. (See Appendix C.1.3 for more examples.) (**b**) For each class, the accuracy of the K test images closest in CLIP embedding space to the most positive/negative SVM captions for that class (averaged over classes). The images closest to the negative caption have a lower accuracy than those closest to the positive caption.

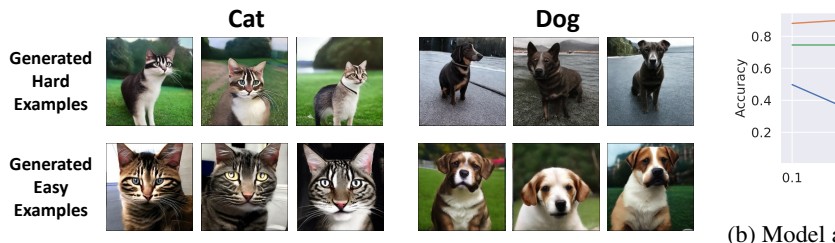

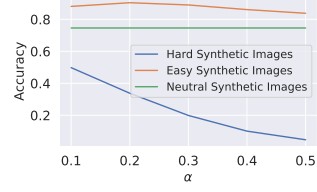

(a) Examples of hard and easy generated images.

(b) Model accuracies on the generated images.

Figure 8: To directly decode the SVM direction into images, we spherically interpolate the extracted direction and the embedding of the reference caption before passing this vector to a diffusion model. (**a**) Examples of synthetic hard and easy images of CIFAR-10 cats and dogs. The generated images match the trends found in Figure 7a. Further examples can be found in Appendix C.1.4. (**b**) Base model accuracy on generated hard and easy images (100 images per CIFAR-10 class) over varying degrees of spherical interpolation ($\alpha$). Neutral images were generated using the reference caption. The base model performs worst on the hard generated images and best on the easy ones.

consider a weaker base model trained with 20% of the original CIFAR-10 dataset, where the other 20% is used for validation and the last 60% is used as extra data for the subset intervention. (See Appendix C.1 for additional experiments, including applying our framework to the larger CIFAR-10 dataset.)

**Finding interpretable subpopulations within the CIFAR-10 dataset.** Figure 7a displays examples of the failure modes identified by our framework. We identify white cats on grass and black dogs as hard, while classifying brown cats and white/brown dogs that are inside as easy.

Do these directions in latent space map to real failure modes of the base model? Without manual annotations, we can no longer directly report the original model's accuracy on these minority groups. However, as a proxy, we evaluate images that are closest in cosine similarity to the captions chosen by the SVM. For example, to get white cats on the grass, we rank the CIFAR-10 cats by how close their embeddings are to the caption "a photo of a white cat on grass" in CLIP space[3]. Using this proxy, in Figure 7b we confirm that the surfaced SVM captions represent real failure modes in the dataset.

**Decoding the SVM direction to generate challenging images.** Since the SVM operates in CLIP space, we can directly decode the extracted direction into an image using an off-the-shelf diffusion model (Blattmann et al., 2022) which also samples from this space. By spherically interpolating between the embedding of the reference caption (e.g., "a photo of a cat") and the extracted SVM direction, we can generate harder or easier images corresponding to the extracted failure mode (Figure 8a). For example, interpolating with the hard direction for CIFAR-10 cats generates white cats on grass, matching the extracted SVM caption in Figure 7a. Across classes, we verify that the original model performs worse on the "hard" generated images and better on the "easy" ones (Figure 8b).

---

[3]In the Appendix C.1, we validate that this approach surfaces images that visually match the caption.

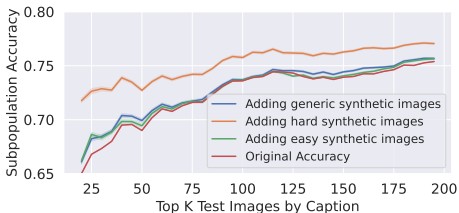 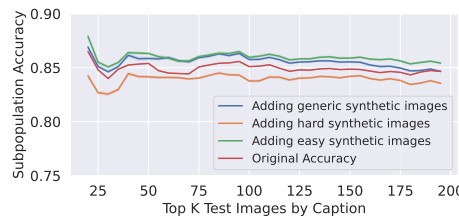

(a) Evaluating images closest to the hard subpopulation    (b) Evaluating images closest to the easy subpopulation

Figure 9: We fine-tune the model using 100 synthetic images per class generated via stable diffusion based on the hard, easy, or reference caption. We measure the accuracy of the K test images closest to the (a) hard or (b) easy SVM caption in CLIP space. Fine-tuning the model on the images generated from the hard captions boosts the accuracy of the model on these corresponding, real test images more than augmenting with generic synthetic images does. The analogous phenomenon does not hold, however, when we target the easy subpopulation.

**Targeted synthetic data augmentation.** By leveraging text-to-image generative models, we can generate images for *targeted* data augmentation. Using a off-the-shelf stable diffusion (Rombach et al., 2022) model, we generate 100 images per class using the corresponding negative SVM caption (e.g., "a photo of a white cat on the grass") as the prompt. After adding these images to the training set, we retrain the last layer of the original model. Fine-tuning the model on these synthetic images improves accuracy on the hard subpopulation — defined according to similarity in CLIP space to the negative caption — compared to using generic images generated from the reference caption (Figure 9a).

It turns out that this procedure for targeted synthetic data augmentation is particularly effective for the identified challenging subpopulation. Generating images using the easy caption (e.g., "a photo of a cat inside") does not improve accuracy on images within that subpopulation any more than augmenting with generic synthetic images (Figure 9b).

It is also possible to augment the dataset with images generated directly from the SVM direction, as in Figure 8a, using a diffusion model that samples directly from CLIP space. In doing so, we could skip the intermediate captioning stage entirely. Currently, the diffusion models that operate in CLIP space are either not open-source (DALL-E 2 (Ramesh et al., 2022)) or trained with less data (Blattmann et al., 2022). However, with further progress in CLIP-space diffusion models, this may be a promising mechanism for tailoring data augmentation directly from the SVM direction.

## 4.2 IMAGENET

We also apply our framework on a larger scale: to find challenging subpopulations in ImageNet (Deng et al., 2009) (see Appendix C.2 for further experimental details). In Figure 10a, we display examples of the failure modes captured by our framework with their associated SVM captions. These biases include over-reliance on color (e.g., the coat of a red wolf) or sensitivity to co-occurring objects (e.g the presence of a person holding the fish tench). We include more examples in Appendix C.2.1.

As in Section 4.1, we validate that these directions correspond to real failure modes by evaluating the accuracy of the test images that are closest to the positive or negative SVM caption in CLIP space (Figure 10b). The identified subpopulations indeed have a significant difference in overall accuracy.

## 5 RELATED WORK

**Debiasing/fairness under known biases.** There are many works on optimizing group fairness with respect to known biases (Feldman et al., 2015; Kamishima et al., 2011; Hardt et al., 2016; Zafar et al., 2017; Kusner et al., 2017; Balashankar et al., 2019); see also (Mehrabi et al., 2021; Hashimoto et al., 2018) and the references therein. Outside the fairness literature, de-biasing approaches in machine learning have been proposed for mitigating the effect of known biases during training (Sagawa et al., 2020; Yang et al., 2019; He et al., 2019; Belinkov et al., 2019; Clark et al., 2019).

**Discovery and mitigation of hard subpopulations.** One line of work uses manual exploration to identify pathologies, such as label ambiguity, in ImageNet (Stock & Cisse, 2018; Vasudevan et al.,

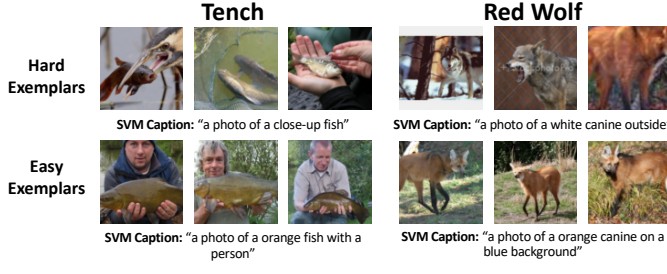
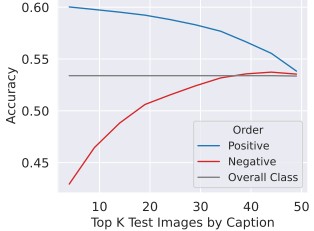

| (a) Most extreme images and extracted SVM captions. | (b) Model accuracies on each identified subpopulation |

Figure 10: (**a**) The most extreme images and captions by SVM decision value for ImageNet classes tench and red wolf. Our framework identifies clear biases on color (e.g red wolf) and co-occurrence of unrelated objects (e.g tench). (**b**) For each class, the accuracy of the top K images that were closest in CLIP space to the most positive/negative SVM captions for that class. The identified hard (negative) subpopulation has a lower accuracy than the identified easy (positive) subpopulation.

2022; Tsipras et al., 2020). Another presents automatic methods for discovering and intervening on hard subpopulations. Sohoni et al. (2020) estimates subclass structure by clustering in the model's latent space, while Liu et al. (2021) directly upweights examples that the model misclassified early in training. Other works directly train a second model which "emphasizes" the examples on which a base model struggles (Nam et al., 2020; Hashimoto et al., 2018; Bao & Barzilay, 2022; Kim et al., 2019; Utama et al., 2020; Sanh et al., 2020). Unlike our framework, these methods focus on the overall test accuracy, rather than finding a consistent failure mode. Perhaps the closest work in spirit to ours is Eyuboglu et al. (2022), which presents an automatic method for identifying challenging subsets by fitting a Gaussian mixture model in CLIP space. However, our SVM provides a stronger inductive bias towards simple subclasses, and furthermore imposes a useful geometry on the latent space for scoring data points, intervening, and selecting captions in a more principled manner.

**Model interpretability.** Model interpretability methods seek to identify the relevant features for a model's prediction. They include gradient- (Simonyan et al., 2013; Dabkowski & Gal, 2017; Sundararajan et al., 2017) and perturbation-based (Ribeiro et al., 2016a; Goyal et al., 2019; Fong & Vedaldi, 2017; Dabkowski & Gal, 2017; Zintgraf et al., 2017; Dhurandhar et al., 2018; Chang et al., 2019; Hendricks et al., 2018; Singla et al., 2021) methods. Other works analyze the model's activations (Bau et al., 2017; Kim et al., 2018; Yeh et al., 2020; Wong et al., 2021). Li & Xu (2021) use a generative model to find biased attributes that impact model predictions. Concurrent work by Abid et al. (2022) uses predefined concept activation vectors to discover latent space perturbations that correct a model's mistake. Finally, several papers investigate the vulnerability of models to specific biases such as sensitivity to texture (Geirhos et al., 2019), background (Xiao et al., 2020), spatial perturbations (Engstrom et al., 2019), common corruptions (Hendrycks & Dietterich, 2019), and non-robust features (Szegedy et al., 2014; Goodfellow et al., 2015; Madry et al., 2018).

# 6 CONCLUSION

In this paper, we introduce a framework for automatically isolating, interpreting, and intervening on hard but interpretable subpopulations of a dataset. In particular, by distilling the model's failure modes as directions in a latent space, our framework provides a mechanism for studying data-points in the context of the model's prediction rules. We thus can leverage our framework to automatically caption the captured failure mode, quantify the strength of distribution shifts, and intervene to improve the model's performance on minority subgroups. Notably, our method enables the effective, automated application of text-to-image models for synthetic data augmentation. We find that our technique succeeds not only on datasets with planted hard subpopulations, but also on natural and widely-used datasets; as such, we hope it will serve as a useful tool for scalable dataset exploration and debiasing.

Several interesting directions remain for further study. One of them is developing sophisticated methods for caption candidate generation. Moreover, a given class may have multiple important failure modes. Therefore, extending our method to disentangle these distinct failure modes—rather than identifying only the most prominent one—is another promising direction for future work.

## 7 ACKNOWLEDGEMENTS

Work supported in part by the NSF grants CCF-1553428 and CNS-1815221 and Open Philanthropy. This material is based upon work supported by the Defense Advanced Research Projects Agency (DARPA) under Contract No. HR001120C0015. HL is supported by the Fannie and John Hertz Foundation and the National Science Foundation Graduate Research Fellowship under Grant No. 1745302. SJ is supported by the Two Sigma Diversity Fellowship.

Research was sponsored by the United States Air Force Research Laboratory and the United States Air Force Artificial Intelligence Accelerator and was accomplished under Cooperative Agreement Number FA8750-19-2-1000. The views and conclusions contained in this document are those of the authors and should not be interpreted as representing the official policies, either expressed or implied, of the United States Air Force or the U.S. Government. The U.S. Government is authorized to reproduce and distribute reprints for Government purposes notwithstanding any copyright notation herein.

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

# A SETUP DETAILS

## A.1 CODE

Our code is available at https://github.com/MadryLab/failure-directions.

## A.2 FORMAL DESCRIPTION OF METHOD

**Preliminaries** We consider a standard image classification problem with image space $\mathcal{X}$ and label space $\mathcal{Y}$, training data $(x, y) \in \mathcal{D}_{train}$, a held-out validation set $\mathcal{D}_{val}$, and test set $\mathcal{D}_{test}$. Using the training data, we have trained a base model (such as a neural network) $f_{nn} : \mathcal{X} \to \mathcal{Y}$ to predict a label given the input. For convenience, we define the "correctness" function $c : \mathcal{X} \to \{-1, +1\}$, where $c(x) = 1$ if $f_{nn}(x)$ correctly outputs the label associated to $x$, and $c(x) = -1$ otherwise.

A *linear support vector machine (SVM)* is a simple linear classifier, which finds a hyperplane separating the data with the maximum margin. Specifically, given $d$-dimensional embeddings $\{x\}_{i=1}^n$ with labels $\{y\}_{i=1}^n \in \{-1, 1\}^n$, a linear SVM computes coefficients $w \in \mathbb{R}^d$ and intercept $b \in \mathbb{R}$ which optimize the objective:

$$\min_{w,b} \frac{1}{2}||w||_2^2 + C \sum_{i=1}^n \max(0, 1 - y_i(w^T x_i + b))$$

This optimization problem seeks to maximize the margin, while penalizing misclassifications using a soft hinge loss. The parameter $C$, learned via cross-validation, trades off between these two objectives.

Once the parameters $w$ and $b$ have been learned, for a given embedding $x$, we can calculate the decision value:

$$h(x) = w^T x + b$$

This represents the confidence of the SVM, and is proportional to the (signed) distance of $x$ to the hyperplane determined by $w, b$. The prediction of the SVM on $x$ is simply then the sign of $h(x)$, and $w$ is the "direction" of the failure mode.

**Diagnose** We evaluate the strength of the most interpretable error pattern within a class $y$ by quantifying the ability of a per-class SVM $h^y$ to predict validation data errors on that class. Recall that each SVM is trained by cross-validation. We use the average balanced accuracy, $\bar{a}$, across cross validation splits to measure the strength of the underlying spurious correlation. Letting $n_y$ denote the number of instances of class $y$ in a given subset of data, the balanced accuracy is defined by

$$\bar{a} = \frac{1}{|\mathcal{Y}|} \sum_{y \in \mathcal{Y}} \frac{1}{n_y} \sum_{x:(x,y) \in \mathcal{D}_{train}} \mathbb{1}(c(x) = 1).$$

Intuitively, the more interpretable and widespread the error pattern is, the better it can be captured by a linear predictor in latent space. To demonstrate this, on CelebA, we subselect the data to underrepresent the hard demographics ("old" "female" and "young" "male") in the training data to lesser or greater degrees. As was shown in Figure 4, $\bar{a}$ increases as the strength of the spurious correlation increases (i.e. as the proportion of hard demographics decreases). This measure can be used to determine which classes have a clear bias present, for use in the following interpretation and intervention steps.

**Interpret** Once the SVMs have captured any difficult subpopulations, we interpret them by surfacing the most extreme examples in the test set, as well as automatically generating captions for the identified failure modes. These interpretability techniques are only possible due to two key properties of SVMs: (1) they are constrained to capture simple (approximately linearly separable) patterns in the embedding space, and (2) their decision boundary is easily accessible, therefore providing a notion of direction and distance in the embedding space.

- **Most Extreme Examples** Each SVM $h^y$ provides a natural measure of *how* hard an input $x$ is via its distance to the decision boundary, which was validated in Figure 2. Thus, we subselect from $\mathcal{D}_{test}$ the $k$ images within class $y$, $\{x_i^-\}_{i=1}^k$, which minimize $\sum_i h^y(\text{clip}(x_i^-))$, and likewise the $k$ images $\{x_i^+\}_{i=1}^k$ which maximize $\sum_i h^y(\text{clip}(x_i^+))$. By definition, these

images exemplify the decision rules learned by $h^y$. Therefore, $\{x_i^-\}_{i=1}^k$ provide a concise visual summary of "hard" examples, and similarly $\{x_i^+\}_{i=1}^k$ of "easy" examples. As was shown in Figure 3, for class $y =$"old", $\{x_i^-\}_{i=1}^k$ indeed consists of all women, while $\{x_i^+\}_{i=1}^k$ consists of all men.

- **Captioning** To generate a caption for the predicted error-prone subpopulation of a class $y$, we leverage the shared vision-language CLIP embedding space on which the SVMs were trained. For a class $y$, let the base caption $\beta$ be "a photo of a $y$" and its embedding $r = \text{clip}(\beta)$. Given a set of candidate captions $\mathcal{C}$, we aim to select the caption that $h^y$ classifies most strongly, in terms of distance from the decision hyperplane, as an error. Consider a candidate caption $\gamma \in \mathcal{C}$ and its embedding $c = \text{clip}(\gamma)$. Since *directions* have been recently shown to be meaningful in CLIP space (Radford et al., 2021), associate each caption with a *direction* in the CLIP embedding space via $d_c = \frac{c-r}{||c-r||}$. The chosen caption for errors on class $y$ is then $\text{argmax}_c h^y(d_c)$.

**Intervene** After identifying and interpreting the hard per-class trends, we use them to improve accuracy of $f_{nn}$ by retraining. Let $\mathcal{H} = \{(x,y) \in \mathcal{D}_{train} \ s.t. \ h^y(\text{clip}(x)) < 0\}$ denote the set of training examples predicted to be hard. We consider various methods for intervening on $\mathcal{H}$ during retraining.

- **Upweighting** Suppose $f_{nn}$ was originally trained to minimize an objective function

$$\mathcal{L}(\mathcal{D}_{train}) = \sum_{(x,y) \in \mathcal{D}_{train}} \ell(y, f_{nn}(x))$$

We modify this objective by upweighting the flagged examples by some factor $h$:

$$\mathcal{L}'(\mathcal{D}_{train}) = \sum_{(x,y) \in \mathcal{D}_{train}} (h\mathbb{1}(x \in \mathcal{H}) + \mathbb{1}(x \notin \mathcal{H})) \cdot \ell(y, f_{nn}(x))$$

- **Subsetting** If there exists an external pool of examles $\mathcal{D}_{ext}$ that were not used to originally train $f_{nn}$, let $\mathcal{H}_{ext} = \{(x,y) \in \mathcal{D}_{ext} \ s.t. \ h^y(\text{clip}(x)) < 0\}$ and let $\mathcal{D}'_{train} = \mathcal{D}_{train} + \mathcal{H}_{ext}$. Thus, we add to the training set the most useful new examples from $\mathcal{D}_{ext}$. (In practice, we simulate this setting by dividing the original training set into $\mathcal{D}_{train}$ and $\mathcal{D}_{ext}$, prior to training $f_{nn}$ the first time.)

## A.3 MODEL TRAINING DETAILS

We train ResNet18 models and perform hyperparameter search using the held out validation set. Dataset specific experimental details can be found in Sections B and C. For all datasets except ChestX-ray14, we use standard softmax classification with cross-entropy loss.

Our training uses SGD with momentum and weight decay. We further use a triangular cyclic learning rate, where the learning rate linearly increases for a pre-specified number of epochs ("peak epochs") until hitting the base learning rate ("peak LR"), and then linearly decays for the rest of the epochs until reaching zero at the end of training.

For ChestX-ray14, we predict for each condition a binary output with binary cross-entropy loss; the threshold is then chosen to maximize the F1 score on the held out validation set.

We train our models using NVIDIA V100 gpus. We use the FFCV framework (Leclerc et al., 2022) to further speed up training.

## A.4 SVM TRAINING DETAILS

We use the pre-trained CLIP (Radford et al., 2021) model, equivalent to a ViT-B/32. We embed the images using this encoder. We whiten the data (using the mean/standard deviation of the embeddings computed from the training images) and then normalize the embeddings to be unit vectors.

We train a linear SVM on the embeddings of the held-out validation set with balanced class weighting to predict whether the model got the image correct or incorrect. The regularization constant is chosen using 2-fold cross validation by selecting the constant with the best balanced accuracy (i.e the average

of the recall obtained for each class). These cross validation scores are also exported to quantify the strength of the captured shift, as described in Section 2.

## A.5 FILTERING DETAILS

After holding out a pool of data that was not used to train the original model, we select those images with the most negative SVM decision values to add to the training set; we also compare to alternative selection methods (by confidence value, and random). We applied this method to CIFAR-100, and reported the accuracies on the hard subpopulation in Figure 6. Below, we report the *overall* test accuracies before and after intervention when adding 100 images per superclass.

|  | Original | Random | Confidences | SVM |
|---|---|---|---|---|
| Overall accuracy | 0.696 | 0.712 | 0.716 | 0.720 |

## A.6 CAPTION GENERATION

In Section 2, we described how we can assign a caption to each captured failure mode by choosing the caption that is most aligned with the positive or negative direction extracted by the SVM. To perform caption assignment, we need to create a set of relevant candidate captions that we expect, roughly, to lie in the same space as the images that the SVM was trained on. In particular, completely nonsensical captions may have undefined CLIP embeddings. Similarly, completely irrelevant captions may be out of distribution for the SVM, which was trained on images of a specific class.

To generate a sensible candidate set, we programmatically generate captions of the form "A photo of a <adjective> <noun> <prepositional phrase>", where the adjective and prepositional phrase are optional. We describe the list of adjectives, nouns, and prepositional phrases for each dataset in the experimental details section for that task. We chose this approach in order to guarantee the descriptiveness of the candidate captions set. Automatically curating this candidate set, for example by generating reasonable sentences using a language model, is an interesting avenue for future work.

## A.7 DIRECTLY DECODING SVM DIRECTIONS WITH DIFFUSION MODELS

In Section 2, we described how to use diffusion models that sample from CLIP space to directly decode our extracted SVM direction. We provide further details here.

Several models, including retrieval augmented diffusion models (Blattmann et al., 2022) and DALL-E 2 (Ramesh et al., 2022) sample from CLIP's shared vision/language space. In contrast, Stable Diffusion (Rombach et al., 2022) uses the hidden states of the CLIP's text encoder (before it is projected into the common image/language space). In our work, we use the pre-trained $768 \times 768$ retrieval-augmented diffusion model (RDM) (Blattmann et al., 2022), but only use text prompting (no retrieval step). The checkpoint can be found here.

We fit the SVM on the same CLIP space that the RDM uses (CLIP ViT-L/14) with $\ell_2$ normalization but not whitening of mean/std. Let $w$ be the normalized SVM direction. For a reference caption with normalized embedding $r$, we then spherically interpolate $r$ and either $w$ or $-w$ (e.g., $z = \text{slerp}(r, w, \alpha)$) to generate hard or easy images respectively. The degree of rotation $\alpha$ specifies the level of difficulty (we generally choose $\alpha$ in the range $[0, 0.3]$). Unless otherwise stated, we use $\alpha = 0.1$ to generate the images. We then pass the spherically interpolated vector in lieu of the text-conditioning vector to the RDM.

## A.8    TAILORED DATA AUGMENTATION USING TEXT-TO-IMAGE GENERATIVE MODELS

In Section 2, we described how we use text-to-image generative models to perform stable diffusion. Specifically, we input the captions extracted by our SVM (easy, hard, and reference) into a pre-trained Stable Diffusion (Rombach et al., 2022) model and generate 100 synthetic images per class. The checkpoint ("sd-v1-4.ckpt") can be found here. We then fine-tune the original model (retraining only the last layer) on the combination of the new images and the original training dataset. Below we report the overall test accuracies after fine-tuning: overall accuracies are maintained up to 1.3%.

|  | Original | Easy | Hard | Neutral |
|---|---|---|---|---|
| Overall accuracy | 0.791 | 0.785 | 0.778 | 0.784 |

## B    DATASETS WITH KNOWN CHALLENGING SUBPOPULATIONS.

### B.1    CELEBA

In Section 3, we applied our framework to the task of predicting age on CelebA (Liu et al., 2015) with a planted correlation with gender. Here, in this section, we discuss the training details and additional experiments on that task.

#### B.1.1    EXPERIMENTAL DETAILS

**Dataset construction**    We construct a training dataset with the following demographic breakdown: 6203 each of "old" "female" and "young" "male", and 24812 of "old" "male" and "young" "female". There are thus four times as many faces of old men vs. old women (resp. young women vs. young men). The held-out validation set contains 1795 examples for each of the demographic categories, and we use the original test split of CelebA for our final test set. We consider the full CelebA dataset and a setting where the validation distribution matches the training distribution in Appendix Sections B.1.7 and B.1.4. We consider images with a resolution of $75\times75$.

**Hyperparameters**    We use the following hyperparameters.

| Parameter | Value |
|---:|:---|
| Batch Size | 512 |
| Epochs | 30 |
| Peak LR | 0.02 |
| Momentum | 0.9 |
| Weight Decay | $5 \times 10^{-4}$ |
| Peak Epoch | 2 |

**Caption generation**    We use a reference caption "A photo of a person." We generate prepositional phrases using the attributes provided by the CelebA dataset. The adjectives, nouns, and prepositional phrases are as follows:

- **Adjectives:** [none, 'old', 'young']

- **Nouns:** ['person', 'man', 'woman']

- **Prepositions:** [ none, 'who has stubble', 'who has arched eyebrows', 'who has bags under their eyes', 'who has bangs', 'who has big lips', 'who has a big nose', 'who has black hair', 'who has blond hair', 'who has brown hair', 'who has bushy eyebrows', 'who has a double chin', 'who is wearing eyeglasses', 'who has a goatee', 'who has gray hair', 'who has heavy makeup on', 'who has high cheekbones', 'who has a mouth that is slightly open', 'who has a mustache', 'who has narrow eyes', 'who has no beard', 'who has an oval face', 'who has a pointy nose', 'who has a receding hairline', 'who has rosy cheeks', 'who has sideburns', 'who has a smile', 'who has straight hair', 'who has wavy hair', 'who has earrings', 'who is wearing a hat', 'who has a lipstick on', 'who is wearing a necktie' ]

We thus generate a caption for every combination of adjective, noun, and prepositional phrase.

#### B.1.2    UPWEIGHTING INTERVENTION

Can we use the linear classifiers from our framework to reduce the model's reliance on gender by intervening during training? To examine this, after flagging examples in the training set as members of the minority population, we upweight these examples during the model retraining. As shown in Table 1, this intervention increases performance on the minority subpopulations, and is even comparable in its impact to the "oracle" approach of upweighting all "young men" and "old women". Note that, since the base model has nearly zero training error, it is confident on all training examples. Thus, confidences are not suitable for guiding an analogous intervention.

| | Test Subgroup Accuracy | | | | Overall Accuracy |
| | Minority Subclasses | | Majority Subclasses | | |
| | Young Male | Old Female | Young Female | Old Male | |
|---|---|---|---|---|---|
| Base Model | $0.568 \pm 0.006$ | $0.542 \pm 0.010$ | $0.915 \pm 0.002$ | $0.904 \pm 0.002$ | $0.795 \pm 0.002$ |
| Oracle Upweight | $0.600 \pm 0.007$ | $0.569 \pm 0.010$ | $0.896 \pm 0.004$ | $0.872 \pm 0.004$ | $0.797 \pm 0.002$ |
| SVM Upweight | $0.590 \pm 0.011$ | $0.568 \pm 0.004$ | $0.903 \pm 0.003$ | $0.901 \pm 0.008$ | $0.796 \pm 0.003$ |

Table 1: The performance (mean/std over five runs) of intervening during training by upweighting examples in the CelebA training set. SVM Upweight doubles the loss weight for all training examples that were predicted as incorrect by our framework. Oracle Upweight doubles the loss weight of all minority examples using the dataset's annotations. We find that intervening using the SVM increases the accuracy of the minority subclasses to almost the same extent as the (optimal) oracle intervention.

### B.1.3 GRADATIONS OF SPURIOUSNESS

In Section 3, we applied our framework to a subset of the CelebA dataset, chosen such that the spurious correlation between age and gender was intensified. Specifically, we vary the intensity of that spurious correlation, and demonstrate that the test error of the per-class SVMs reflects the degree of the spurious correlation. In particular, for $n$ from 1 to 8, the train dataset is filtered such that there are $n$-times as many old male instances as young male instances, and $n$-times as many young female instances as old female instances.

As shown in Figure 11, the test accuracy of the base model decreases as the spurious correlation grows stronger, because the base model has relied more and more on the spurious correlation. This complements Figure 4 in the main text, which demonstrated that the SVMs fit the test data better with increasing spurious correlation.

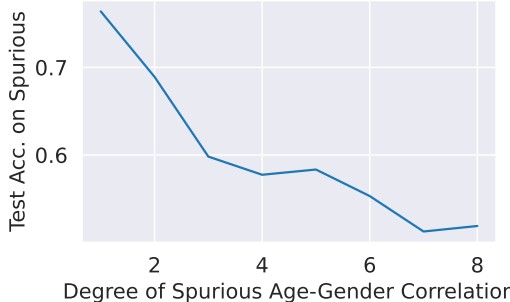

Figure 11: Test accuracy on spurious instances of CelebA, as a function of the spuriousness $n$ of the training dataset.

### B.1.4 CELEBA EXPERIMENTS WHEN THE VALIDATION MATCHES THE TRAINING DISTRIBUTION.

In Section 3, we applied our framework to the CelebA dataset and used a validation set that was balanced across age and gender. Here, we instead consider the case where the validation set matches the train distribution, and thus also contains the strong spurious correlation with gender: specifically, the validation set contains 810 old female and young male faces, and 3240 old male and young female faces.

In this setting, there are fewer minority examples for the SVM to learn from, which thus makes the task of selecting errors harder. However, we find that our framework still captures the gender correlation (Figure 12). Moreover, our method is able to isolate this hard subpopulation better than using model confidences (Figure 13).

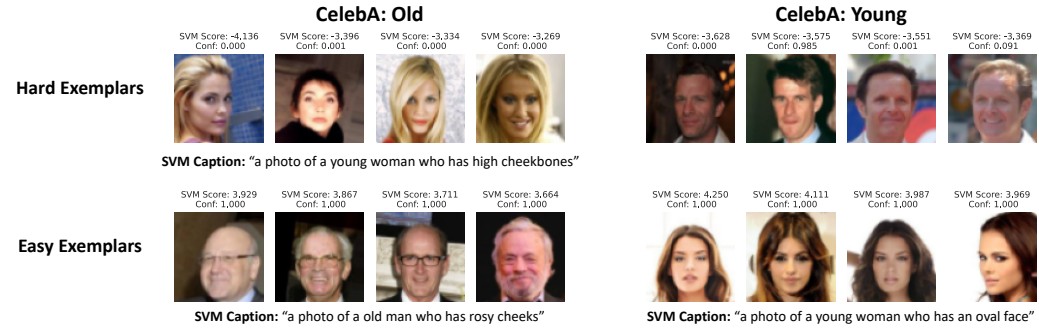

Figure 12: The images and captions for each class with the most extreme SVM decision values

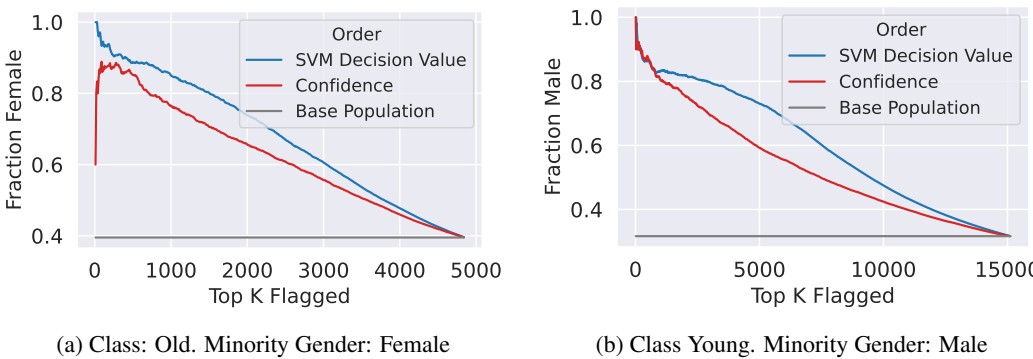

(a) Class: Old. Minority Gender: Female

(b) Class Young. Minority Gender: Male

Figure 13: We repeat our CelebA experiments from Section 3, using a held out validation set that matches the training distribution. We display for each class, the fraction of the top K images that are of the minority gender when ordering the images by either their SVM decision value or by the model's confidences.

### B.1.5 FURTHER EXPERIMENTAL DETAILS ON BASELINES

In Figure 2 of from Section B of the main paper, we compare the efficacy of using our SVM (versus several baselines) for selecting the minority gender in the CelebA setting (replicated again below). We found that our method was able to more consistently select the hard subpopulation than the other baselines. In this section, we discuss the training details for this experiment.

**Domino**    Domino (Eyuboglu et al., 2022) is an automatic method for identifying challenging subsets of the data ("slices") by fitting a Gaussian mixture model in CLIP embedding space. They then choose captions generated from a large language model that are closes to the mean of each identified cluster. Our use of an SVM provides a stronger inductive bias towards simple subclasses. Furthermore by encapsulating the failure modes as a direction in the latent space, we can more easily score, intervene,

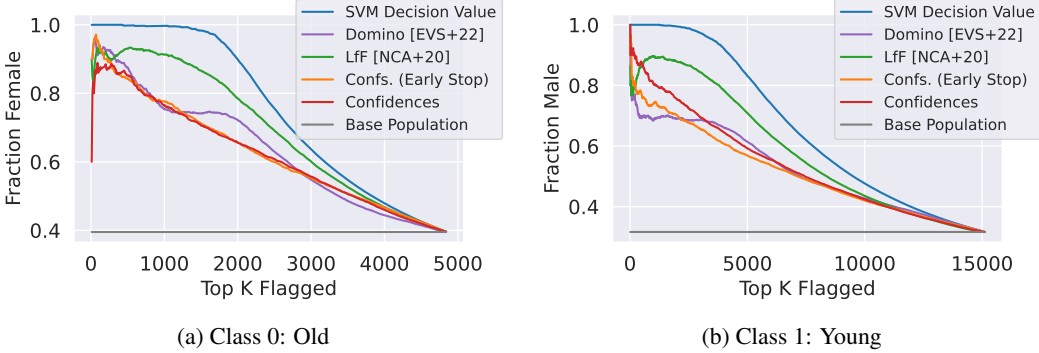

(a) Class 0: Old                    (b) Class 1: Young

Figure 14: Repeated again for convenience: Figure 2 from Section B of the main paper. For each class in CelebA, the fraction of test images that are of the minority gender when ordering the images by either their SVM decision value, the model's confidences, or a variety of other baselines. Our framework more reliably captures the spurious correlation than all other baselines.

and caption the failure modes. In particular, we are able to caption the extracted failure mode directly, rather than using the proxy of a cluster of hard examples. On the other hand, Domino can more easily distinguish between multiple failure modes for the same class because it identifies multiple hard clusters in the latent space.

We use the same parameters as suggested in the paper with *y log likelihood weight*=10, *y hat log likelihood weight*=10, *num slices*=2.

*Captioning:* We first surface the top 3 identified negative captions from our framework and from Domino:

- **Our Framework:**
    - *Old*: "a photo of a young woman who has an oval face", "a photo of a young woman who has heavy makeup on", "a photo of a young woman who has brown hair"
    - *Young*: "a photo of a man who has no beard", "a photo of a man who has a mouth that is slightly open", "a photo of a man who has a smile"

- **Domino**
    - *Slice 0*: "a photo of her debut album.", "a photo of carole lombard.", "a photo of tina turner."
    - *Slice 1*: "a photo of a compelling man.", "a photo of a satisfied man.", "a photo of a hungry man."

Both methods capture gender as the primary failure mode. However, Domino tends toward more specific and thus sometimes more niche captions (i.e "a photo of carole lombard") than overall trends. This is because Domino picks the caption that happens to be closest to the mean of the slice.

**Learning From Failure**   Learning from Failure (LfF) (Nam et al., 2020), uses a biased classifier trained with a Generalized Cross Entropy (GCE) loss to select hard examples. Specifically, if $p(x)$ is the softmax probability vector assigned by the model for input $x$, the GCE loss for example $(x, y)$ is

$$GCE(p(x), y) = \frac{1 - p_y(x)^q}{q}$$

For this baseline, we train our model with GCE loss using the default hyperparameter $q = 0.7$ as from the original LfF paper.

**Early Stopping**   In Just Train Twice (JTT) (Liu et al., 2021), a classifier trained on only a few epochs is used to identify low confidence examples. For this baseline, we stop at epoch=10 (out of 30) and evaluate model confidences.

### B.1.6 USING OTHER LATENT SPACES

In our paper, we distill failure modes using the CLIP latent space. However, our method is not specific to CLIP (we can use any embedding that can capture important features and is agnostic to the original training dataset). In Figure 15, we explore using two other latent spaces for our method:

- **Inception:** We take features from an Inception V3 model (Szegedy et al., 2016) trained on ImageNet.
- **Original Latent Space:** We take features from the model's original latent space (i.e. its penultimate layer).

We find that using our method with Inception embeddings is only slightly worse than using CLIP. Since CLIP embeds both language and images in the same latent space, using CLIP further enables us to automatically caption the extracted directions.

However, since the model's latent space is tied closely to the training dataset, training an SVM on the original latent space closely mirrors model confidences. Moreover, since the original model fit the training data perfectly, the SVM extracted by our method predicts all training examples as correct — even images that belong to the minority subpopulation — and thus cannot be used for upweighting interventions. Thus, our method works best when using embeddings that are agnostic to the specific dataset.

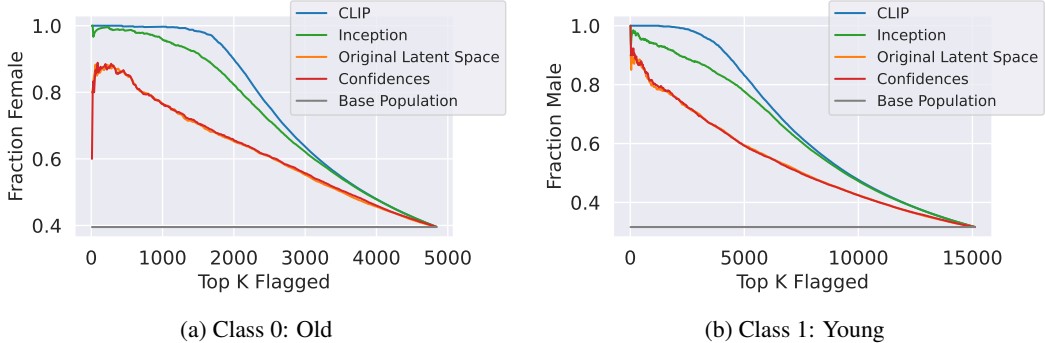

(a) Class 0: Old          (b) Class 1: Young

Figure 15: We repeat our CelebA experiments from Section 3, and compare using the CLIP latent space to using Inception features or the model's original latent space.

### B.1.7 FULL CELEBA DATASET

In Section 3, we subselected images from the CelebA dataset to intensify the spurious correlation between age and gender. Here, we train a base model on the unaltered CelebA dataset instead. Table 2 summarizes the presence of this spurious correlation in the dataset. The per-class SVMs pick up on the spurious correlation, and to a greater extent than do confidences, as shown in Figures 16a and 16b. The validation set on which the SVMs were trained is the standard CelebA validation set.

| | Class (Age) | |
| --- | --- | --- |
| % Minority Gender | Old | Young |
| Overall | 39.5% | 31.6% |
| Incorrect | 55.0% | 57.1% |

Table 2: **First row:** the percentage of each age group consisting of that age group's minority gender (female for old, male for young). **Second row:** the percentage of the base model's mistakes on each age group which are of the minority gender. Together, these capture the degree of spurious correlation present, as the base model makes a disproportionately high number of mistakes on the minority gender for each age group.

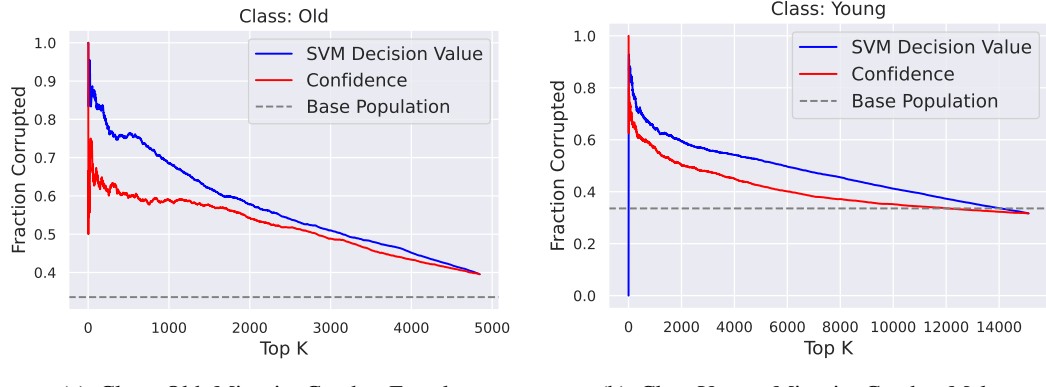

(a) Class: Old. Minority Gender: Female          (b) Class Young. Minority Gender: Male

Figure 16: For each class in the original CelebA dataset, the fraction of the top K images that are of the minority gender when ordering the images by either their SVM decision value or by the model's confidences.

## B.2    COLORED MNIST

In the Colored MNIST dataset (Arjovsky et al., 2019), a spurious correlation is introduced between tint and label for the handwritten MNIST dataset. The learning task labels $y$ are binary, and are generated as follows. We begin with labels $\tilde{y}$, which are 0 for digits less than 5 and 1 for digits greater than or equal to 5. To obtain the labels $y$ actually used in the classification task, the labels $\tilde{y}$ are flipped with probability 0.25. In other words, 75% of the time, the class label $y$ in the learning task is associated with the digit, and this feature generalizes to the test set.

However, we use color to plant a spurious correlation between the class label $y$ and the *tint* of the digit. In particular, we add a red/green tint that correlates perfectly with $y$ ($y = 0$ corresponding to red, and $y = 1$ corresponding to color green), to 90% of the training set, but only 50% of the validation and set tests. As a result, there is a spurious correlation between the tint and the label $y$ in the train set, which does not generalize to the validation and test sets.

Thus on the training data, the color of the digit is a better predictor of its label $y$ than the digit's true shape, but the digit's true shape is a better predictor than its color on the test and validation data.

**Hyperparameters.**    We use the following hyperparameters for training.

| Parameter | Value |
|---:|:---|
| Batch Size | 512 |
| Epochs | 15 |
| Peak LR | 0.1 |
| Momentum | 0.9 |
| Weight Decay | 5e-04 |
| Peak Epoch | 2 |

As shown in Figure 17, the fit of the SVM on the test data improves as the spurious correlation grows stronger. For each value of $x$, we also intervene by doubling the weight of the examples flagged as "hard" by the SVM. Figure 18 demonstrates that while the test accuracy decreases with increasing spurious correlation, intervention is effective (and increasingly so with $x$) at improving test accuracy.

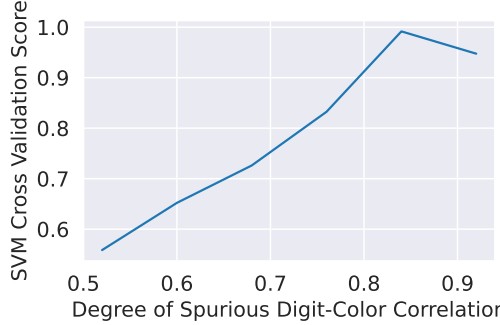

Figure 17:  Cross-validation score on the test set as a function of the spuriousness, i.e. the tinted fraction $x$ of the training set, for Colored MNIST.

Figure 18:   Test accuracy of Colored MNIST before and after the upweighting intervention, as a function of the spuriousness of the training set.

### B.3 IMAGENET-C

#### B.3.1 EXPERIMENTAL DETAILS

ImageNet-C is a dataset created by Hendrycks & Dietterich (2019) to benchmark neural nets' robustness to common perturbations. Using the associated code, we create a corrupted ImageNet dataset in which 10% of the training set is corrupted, and 50% of the validation and test sets are corrupted. For simplicity, we restrict corruptions to Gaussian pixel-wise noise, with variance equal to 0.26. We train a ResNet18 as the base architecture, with standard ImageNet normalization. The resultant validation and test accuracies are 51.1% and 48.9%, respectively, but drop to 45.5% and 43.0% if restricted to corrupted images (validating that the corrupted images are indeed harder).

**Hyperparameters.** Hyperparameters were chosen to match those of the ordinary ImageNet dataset:

| Parameter | Value |
|---|---|
| Batch Size | 1024 |
| Epochs | 16 |
| Peak LR | 0.5 |
| Momentum | 0.9 |
| Weight Decay | 5e-4 |
| Peak Epoch | 2 |

#### B.3.2 RESULTS

**Capturing common corruptions.** We find that the per-class SVMs pick up on the Gaussian corruption, as shown in Figure 19. Since the corruption is light, this is non-trivial: in particular, notice that the images chosen as "hardest" are not necessarily corrupted, as shown in Figure 20.

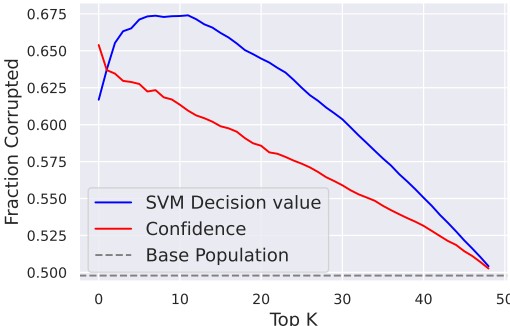

Figure 19: The fraction of ImageNet-C test images that are corrupted with Gaussian noise when ordering the images by either their SVM decision value or by the model's confidences, averaged over all classes. Our framework more reliably captures the corruption than using confidences.

**Intervention.**    In this setting, it does not make sense to intervene based on the captured corruption: since blurred images represent neither a spurious correlation nor an underrepresented subpopulation, but instead simply a "hard" class, we expect neither upweighting nor subsetting to improve performance. The purpose of this dataset is to exhibit that our method can *capture* the hard subclass of corrupted images, even though (as shown in Figure 20) there are many ways an image can be hard, because it is a *shared* pattern across many of the errors.

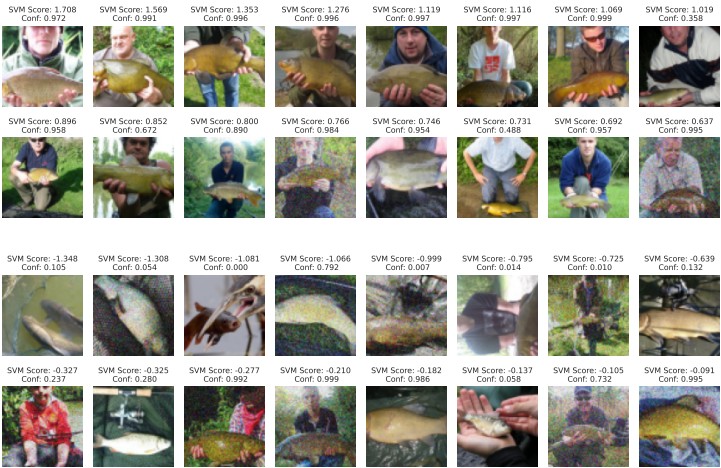

Figure 20: The images with the most positive (upper) and negative (lower) decision values for the ImageNet-C class "tench". While more of the bottom images are blurred, the "hardest" images may be tenches in strange positions.

### B.4 CIFAR-100

In this section, we explain the experimental details of the experiment on CIFAR-100 in Section 3. Specifically, we consider the task of predicted the super-classes of the CIFAR-100 dataset, which contains 20 super-classes (each of which contain 5 subclasses).

**Dataset construction**   We split the original training split into 20% validation set, 40% training set, and 40% extra data (used in the subset intervention). The original test split serves as our test set. In the training dataset, we choose one subclass to under-represent by removing 75% of that subclass from the data. We choose this subclass by picking the worst-performing subclass for each superclass based on a model trained on the original training split. Specifically, we drop the classes:

- {aquatic mammals: beaver, fish: shark, flowers: orchid, food containers: bowl, fruit and vegetables: mushroom, household electrical devices: lamp, household furniture: table, insects: caterpillar, large carnivores: bear, large outdoor man-made things: bridge, large outdoor natural scenes: forest, large omnivores and herbivores: kangaroo, medium sized mammals: possum, non-insect invertebrates: lobster, people: baby, reptiles: lizard, small mammals: squirrel, trees: willow tree, vehicles 1: bus, vehicles 2: streetcar}

**Hyperparameters.**   We use the following hyperparameters to train our models.

| Parameter | Value |
|---:|:---|
| Batch Size | 512 |
| Epochs | 35 |
| Peak LR | 0.5 |
| Momentum | 0.9 |
| Weight Decay | $5.0 \times 10^{-4}$ |
| Peak Epoch | 5 |

#### B.4.1 RESULTS ON 80% TRAIN, 20% VALIDATION

In this section, we discuss results on a CIFAR-100 setup with 80% of the original train split allocated for the train dataset and 20% allocated for the validation dataset (and no extra data). As above, in the training set, we underrepresent one subclass per superclass (by removing 75% of that subclass from the dataset). We then will explore the downstream implications of the CV score in this setting.

**Efficacy in isolating the minority subpopulation**   In Figure 21, we replicate the experiment in Figure 5 for this new split. The results are similar: using the SVM's decision value more consistently identifies the top K images over using model confidences.

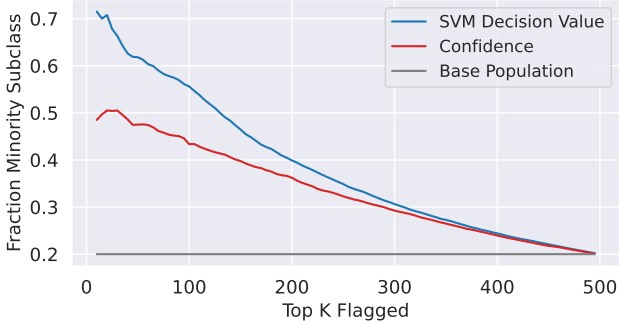

Figure 21: We replicate the experiment in Figure 5 for a scenario where 80% of the training dataset is train and 20% is validation. For each superclass, the fraction of the top K flagged test images that belong to the underrepresented subclass (averaged over classes) is shown. We compare using the SVM's decision value and the model's confidences to select the top K images; the SVM more consistently identifies the minority subpopulation.

**A deep-dive into the cross-validation (CV) score** Recall from Section 2 that the cross-validation (CV) score of the SVM serves as a measure of the extracted failure mode's strength in the dataset. In Figure 4 of the main paper, we found that (in the CelebA setting) the CV score of the SVM was highly correlated with the degree of the planted shift.

In this section, we analyze the downstream implications of the CV score in the CIFAR-100 setup. Specifically, we evaluate our method on CIFAR-100, where the minority subpopulation is underrepresented to different degrees (e.g., removing 10%, 20%, etc of the minority subclass).

As in the CelebA case, the mean CV score across all classes increases as we remove more of the minority subclass (Figure 22a). The CV score is negatively correlated with the downstream model's accuracy on the minority subpopulation (Figure 22b). Thus, the CV score can help measure the degree of the underlying failure mode and its impact on downstream performance.

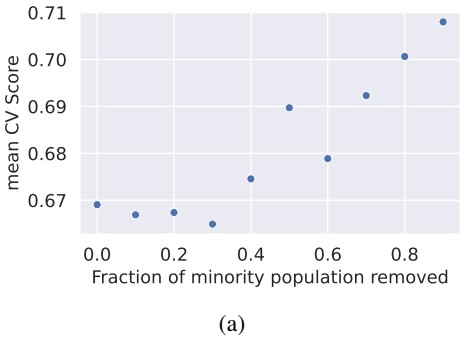
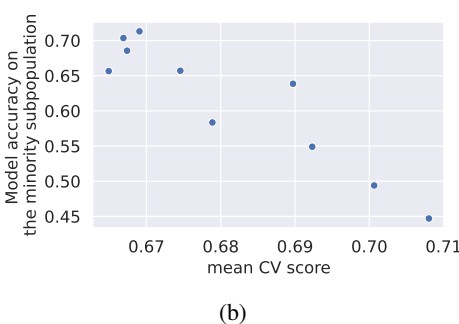

(a)                                         (b)

Figure 22: CV score is correlated with the strength of the failure mode. **(a)** CV score (averaged over classes) when different fractions of the minority population have been removed from the training data. CV score increases as the degree of underrepresentation becomes more severe. **(b)** The CV score negatively correlates with the model's final accuracy on the minority subpopulation.

Since the CV score is correlated with the strength of the failure mode, it also reflects the membership of the error set (predicted and actual). With higher CV scores, more of the predicted errors are actually part of the planted subpopulation (Figure 23).

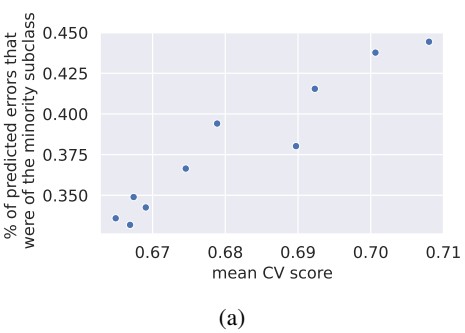
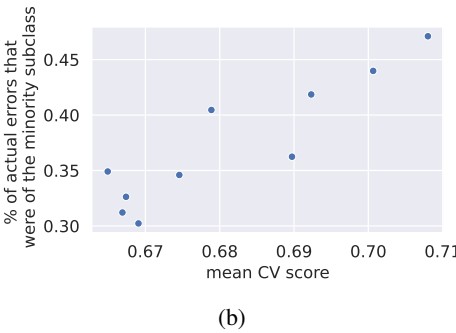

(a)                                         (b)

Figure 23: As shown in Figure 22a, as the degree of the underrepresentation goes up, the CV score also increases. The CV score is thus correlated with the proportion of the error set — **(a)** predicted by the SVM and **(b)** actual — that is actually part of the minority subpopulation.

**Inter-class CV Scores** We now fix the degree of underrepresentation (removing 75% of the minority examples). In this case, the size of the minority subclass is the same for each of the 20 superclasses. However, the impact of underrepresentation is not equal between classes. For example, one might need fewer examples to distinguish a beaver vs. a bear over two different types of trees. Recall that we train an SVM per class: thus, each class has its own CV score. What do the differences in the class CV score signify?

For each of the 20 superclasses, we plot the CV score of that class's SVM against the fraction of errors (predicted and actual) which are of the minority subpopulation (Figure 24). Indeed, classes with a higher CV score seem to indicate a stronger shift, in that a greater proportion of error set belongs to the minority subpopulation.

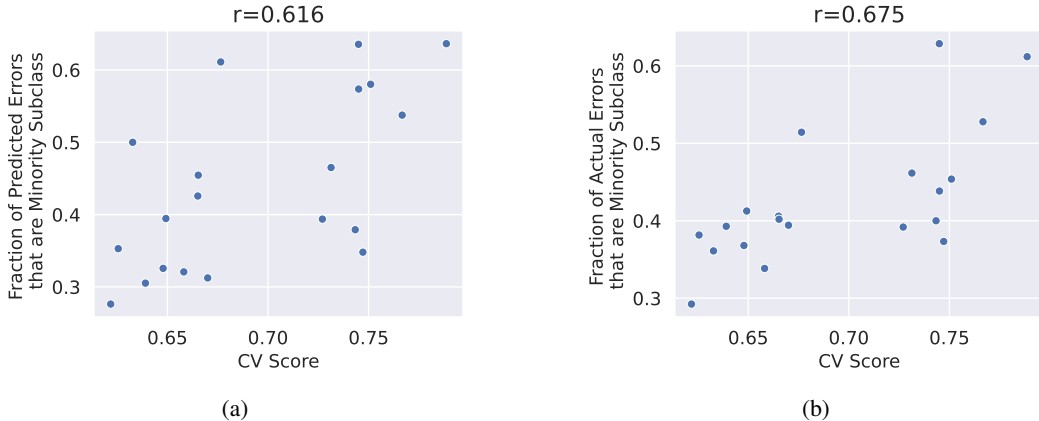

(a)                                                          (b)

Figure 24: For each of the 20 superclasses, we plot the CV score for that class against the fraction of the error set for that class — **(a)** predicted by the SVM and **(b)** actual — that belongs to the minority subpopulation. We further display the Pearson correlation.

## C    NATURAL DATASETS

### C.1    CIFAR-10

In this section, we describe the experimental details and additional results for applying our framework on the CIFAR-10 dataset.

#### C.1.1    EXPERIMENTAL DETAILS

**Dataset.**    Since the original CIFAR-10 model has very high (93%) accuracy, we do the majority of our experiments using a subset of CIFAR-10. Specifically, we train the model with 20% of the original training split, and reserve 20% for the validation set and 60% as extra data for the subset intervention. We discuss results on the full CIFAR-10 dataset in Appendix Section C.1.7.

**Hyperparameters**    We use the following hyperparameters.

| Parameter | Value |
|---:|:---|
| Batch Size | 512 |
| Epochs | 35 |
| Peak LR | 0.5 |
| Momentum | 0.9 |
| Weight Decay | $5.0 \times 10^{-4}$ |
| Peak Epoch | 5 |

For fine-tuning, we use the same parameters, except with peak LR of 0.1 and for 15 epochs.

#### C.1.2    CAPTION GENERATION DETAILS

For our CIFAR-10 experiments, we consider two caption sets: CIFAR-Simple, and CIFAR-Extended. The two caption sets only differ in their sets of possible nouns (extended has a larger vocabulary): they otherwise have the same list of adjectives and prepositions. In the main paper, and unless otherwise noted, we use CIFAR-Simple: this is because the diffusion models struggle with extended vocabularies. In the appendix, we include results with CIFAR-Extended where noted.

For each CIFAR-10 class, we use the reference caption "a photo of a <class>." We then generate a candidate caption set of the form "a photo of a <adjective> <noun> <preposition>". The adjectives, nouns, and prepositional phrases are selected as follows:

- *Adjectives*: [none, 'white', 'blue', 'red', 'green', 'black', 'yellow', 'orange', 'brown'],
- *Prepositions*: We choose a set of prepositions for each class. Specifically, all classes have the following as potential prepositions: [None, 'outside', 'inside', 'on a black background', 'on a white background', 'on a green background', 'on a blue background']. Then each class has the following class-specific prepositions:
    - 'dog': ['on the grass', 'in a house', 'in the snow', 'in the forest'],
    - 'cat': ['on the grass', 'in a house', 'in the snow', 'in the forest'],
    - 'bird': ['flying', 'perched', 'in the air', 'on the ground'],
    - 'horse': ['in a field', 'in the grass'],
    - 'airplane': ['flying', 'in the air', 'on the tarmac', 'on a road'],
    - 'truck': ['on the road', 'parked'],
    - 'automobile': ['on the road', 'parked'],
    - 'deer': ['in a field', 'in the grass', 'in the snow', 'in a forest'],
    - 'ship': ['in the ocean', 'docked', 'in the water', 'on the horizon'],
    - 'frog': ['in the grass', 'in a pond'],

**Nouns**: For CIFAR-Simple, we do not vary the noun (i.e., captions of a class cat will use the noun cat). For CIFAR-Extended, we find the corresponding synset in the WordNet (Miller, 1995) hierarchy. We then choose as possible nouns all the common names of the synsets that are descendents of the

CIFAR synset. For example, for the class cat, a possible noun is "Persian cat." We exclude nouns that contain words that are not in the NLTK (Loper & Bird, 2002) words corpus.

For both sets, we generate the captions by taking every combination of adjective, noun, verb.

### C.1.3 MOST EXTREME EXAMPLES AND CAPTIONS

In Figure 25 we show the most extreme examples and captions by SVM decision value for all 10 classes.

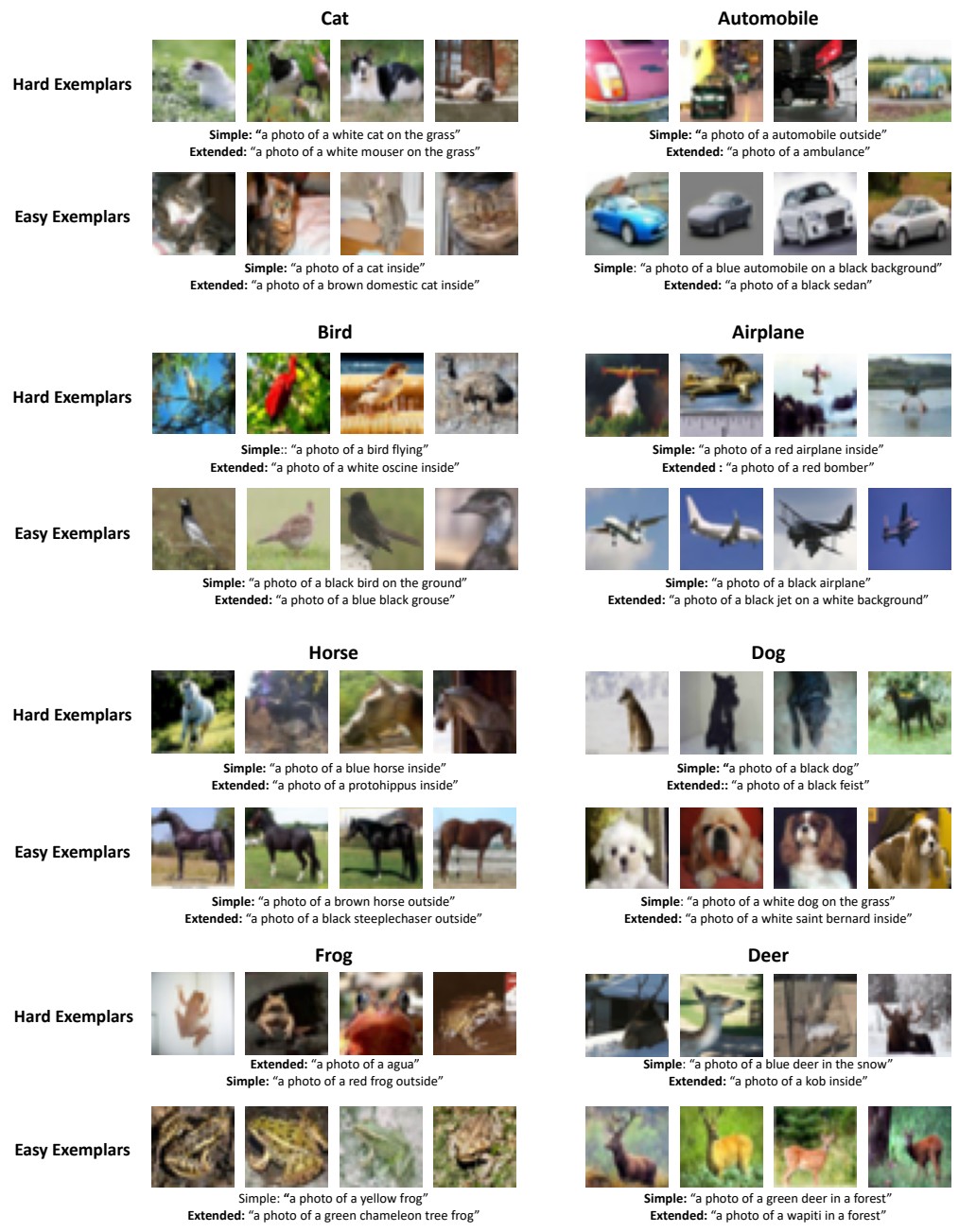

Figure 25: Most extreme images with captions from CIFAR-Simple or CIFAR-Extended

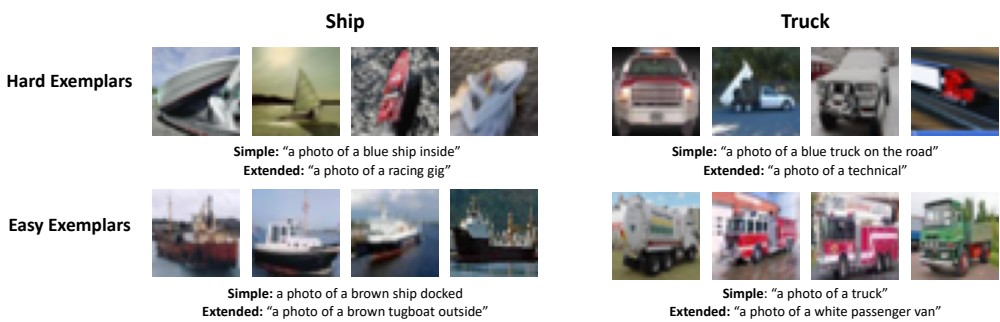

Figure 25: Most extreme images with captions from CIFAR-Simple or CIFAR-Extended

### C.1.4 MORE EXAMPLES OF DIRECTLY DECODING SVM DIRECTION

In Figure 26, we include more examples of using spherical interpolation to directly decode the SVM direction for CIFAR-10 (as in Figure 8a).

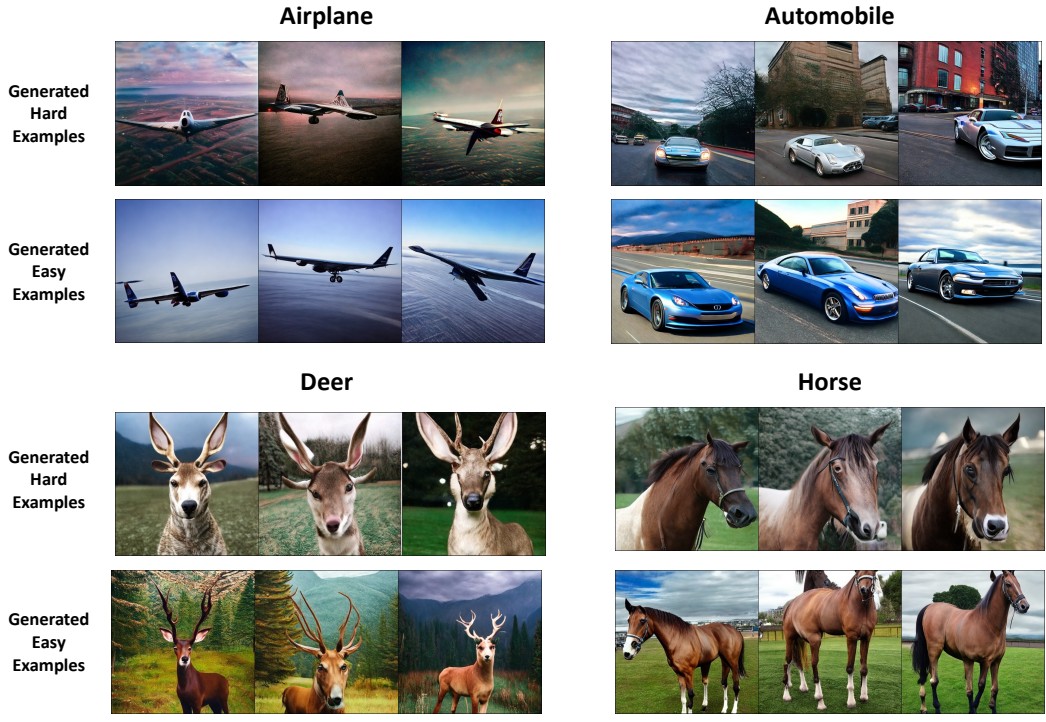

Figure 26: More images generated by directly decoding the SVM direction.

### C.1.5 VALIDATING THE PROXY OF EVALUATING IMAGES CLOSEST TO THE CAPTIONS

In Figure 7b from the main paper, we used the K closest images to the positive or negative SVM caption as a proxy for the subpopulation that corresponded to that caption. Here, we validate that the images which are closest in cosine distance to a given caption indeed closely match the description of the caption itself. We use the CIFAR-Extended caption set.

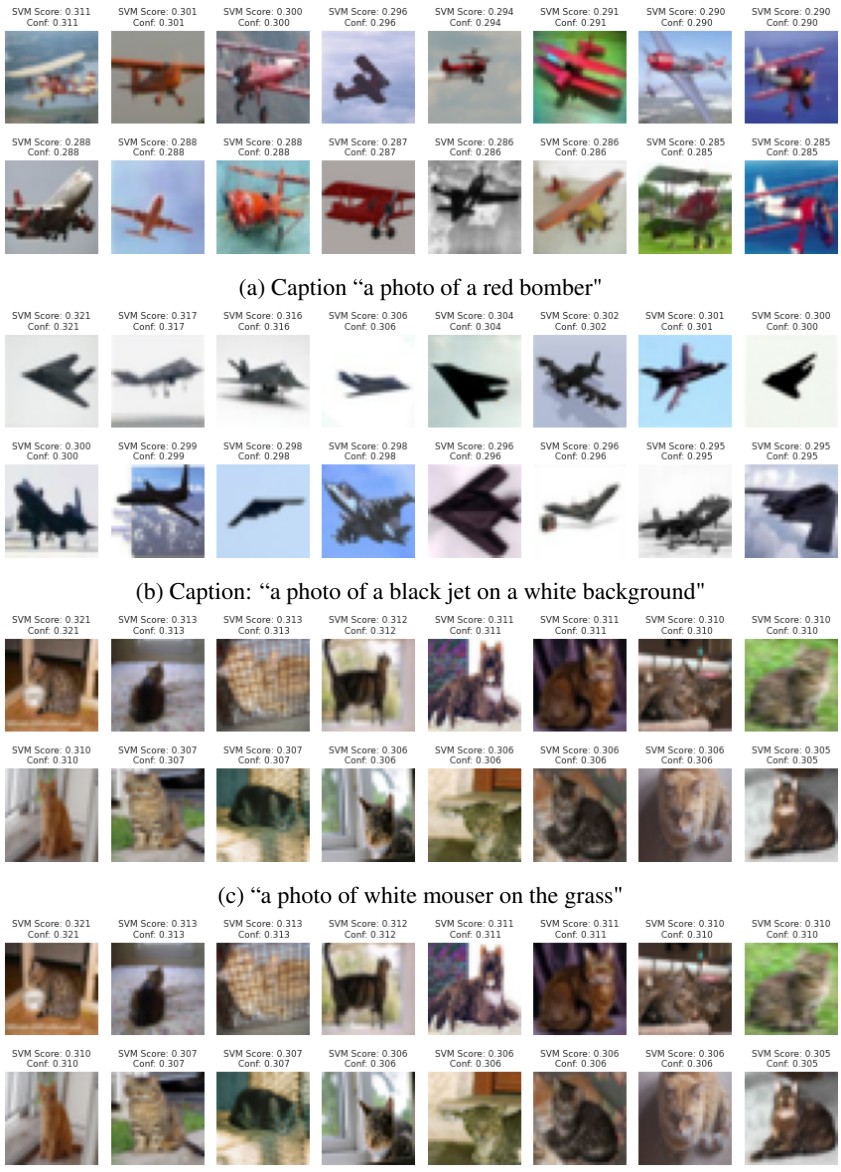

(a) Caption "a photo of a red bomber"

(b) Caption: "a photo of a black jet on a white background"

(c) "a photo of white mouser on the grass"

(d) "a photo of a brown domestic cat inside"

Figure 27: The images closest in cosine distance to the given CLIP caption.

### C.1.6 FILTERING INTERVENTION FOR CIFAR-10

Here, we perform the filtering intervention for CIFAR-10. For each class, we take the top 100 examples according to our SVM decision values, the model's base confidence, or a random baseline, and then add those examples to the training dataset (Figure 28). Without annotations, we again use the proxy of evaluating the images closest to the extracted negative SVM captions (here from CIFAR-Extended). We find that our framework improves the performance on the minority subgroups to a larger extent than other methods.

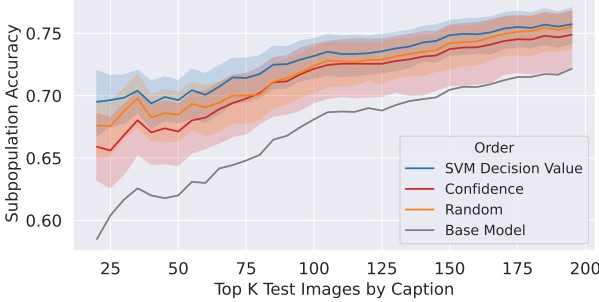

Figure 28: For each CIFAR-10 class, the accuracy of the K examples closest to the negative SVM caption after adding 100 images from the extra data (and retraining) either at random, based on the SVM decision values, or based on the model confidences. Choosing these 100 images by using the SVM decision value best improves the accuracy population described by the negative caption. Results are averaged over classes and reported over 10 runs.

### C.1.7 RESULTS FOR FULL CIFAR-10 DATASET

Here we show the results of applying our framework on the full CIFAR-10 dataset. We therefore use the entire training split, except for 20% used for validation. We find that similar failure modes are captured (Figure 29), and the identified negative subpopulations have a lower accuracy than the identified positive subpopulations (Figure 30).

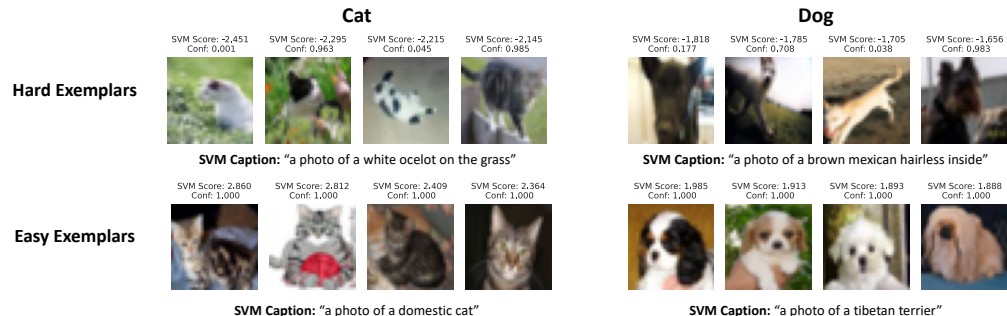

Figure 29: Most extreme images/captions surfaced by our framework for the full CIFAR-10 dataset

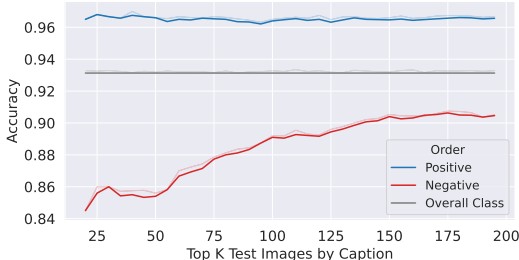

Figure 30: Averaging over 10 classes, we evaluate the accuracy of the top K images closest in cosine distance to the positive or negative caption on the test set.

## C.2 IMAGENET

We discuss the experimental details for the ImageNet dataset.

**Dataset.** We consider 40% of the original dataset as training data, 40% as extra data, and the last 20% as the validation set. We perform training and evaluation at a resolution of 224×224.

**Hyperparameters** We use the following hyperparameters, which were taken from the recommended FFCV configuration:

| Parameter | Value |
| --- | --- |
| Batch Size | 1024 |
| Epochs | 16 |
| Peak LR | 0.5 |
| Momentum | 0.9 |
| Weight Decay | $5 \times 10^{-4}$ |
| Peak Epoch | 2 |

**Caption generation** We again generate the captions programmatically using the caption "a photo of a <adjective> <noun> <prepositional phrase>." We use the following adjectives: [None, 'white', 'blue', 'red', 'green', 'black', 'yellow', 'orange', 'brown', 'group of', 'close-up', 'blurry', 'far away'].

Since there are 1000 ImageNet classes, we need a way to group classes together to more scalably assign prepositional phrases that makes sense (for example, the sentence "a purple lawn-mower that is flying in the air" is likely out of distribution for the CLIP encoder.) We consider a set of ancestor nodes in the WordNet hierarchy, and assign each ImageNet class to its closest ancestor in the set. The set of ancestors and the number of ImageNet classes associated to each is below.

- {'fish': 16, 'bird': 59, 'amphibian': 9, 'reptile': 36, 'invertebrate': 61, 'mammal': 30, 'marsupial': 3, 'aquatic mammal': 4, 'canine': 130, 'feline': 13, 'rodent': 6, 'swine': 3, 'bovid': 9, 'primate': 20, 'device': 124, 'entity': 59, 'vehicle': 7, 'aircraft': 4, 'structure': 57, 'wheeled vehicle': 40, 'container': 32, 'equipment': 37, 'implement': 36, 'covering': 43, 'furniture': 21, 'vessel': 32, 'fabric': 6, 'train': 1, 'instrumentality': 12, 'appliance': 12, 'bus': 3, 'food': 38, 'fruit': 16, 'geological formation': 9, 'person': 3, 'flower': 2, 'fungus': 7 }

We then use the following set of prepositions for each class. Specifically, all classes use the set of prepositions [None, 'outside', 'inside', 'on a black background', 'on a white background', 'on a green background', 'on a blue background', 'on a brown background']. Furthermore, depending on their assigned ancestor class, they use the following class specific prepositions

- 'reptile', 'amphibian': ['in a tank', 'on the ground', 'on a rock', 'in the grass']
- 'marsupial', 'swine', 'bovid', 'feline', 'canine', 'rodent': ['in the grass', 'in a house', 'in the forest', 'on the ground', 'with a person']
- 'fungus': ['on the ground', 'in the grass']
- 'food': ['on a plate', 'one the ground', 'on a table', 'with a person']
- 'geological formation', 'train', 'mammal', 'structure', 'entity': []
- 'wheeled vehicle', 'bus', 'vehicle: ['on the road', 'parked']
- 'fruit', 'fruit', 'flower': ['on a table', 'on the ground', 'on a tree', 'with a person', 'in the grass']
- 'fish', 'aquatic mammal': ['in a tank', 'with a person', 'underwater']
- 'person': ['in a house', 'on a field']
- 'equipment','instrumentality', 'appliance', 'container', 'fabric', 'covering', 'device', 'implement: ['on a table', 'with a person', 'with a hand']
- 'primate': ['in a tree', 'on the ground', 'in the grass']
- 'invertebrate': ['in the grass', 'on the ground', 'in a house']

- 'bird': ['in the air', 'on the ground', 'in a cage', 'in the grass', 'flying', 'perched']
- 'aircraft': ['in the air', 'on the ground']
- 'furniture': ['in a house']
- 'vessel': ['in the water', 'docked', 'in the ocean']

Finally, the ImageNet class names themselves can include quite niche words (which may not be well-represented by CLIP). We thus use as the noun the ancestor corresponding to the ImageNet class. As a reference caption, we use "A photo of a <ancestor>".

### C.2.1 Further ImageNet Examples

In Figure 31 we show additional results for Imagenet classes cauliflower and agama.

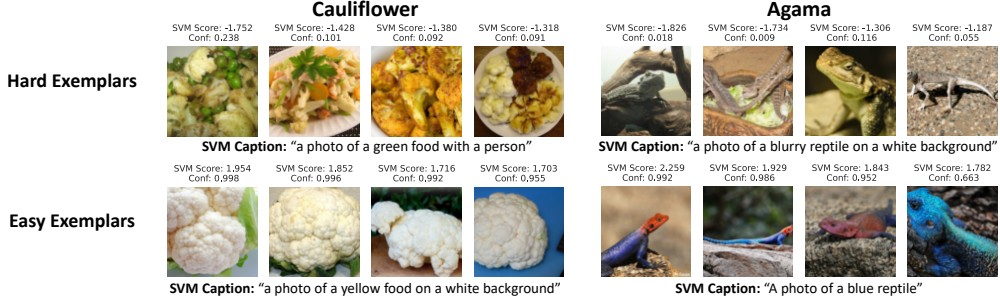

Figure 31: Most extreme images/captions surfaced by our framework for ImageNet classes cauliflower and Agama

### C.2.2 ImageNet and CIFAR-10 normalized plots

Since ImageNet and CIFAR-10 have different test set sizes (50 examples and 1000 examples per class respectively), here we plot equivalents of Figures 7b and 10b. In Figure 32, instead of plotting the top K images considered on the x-axis, we plot the fraction of the test set considered (from to 0% to 100% of the test set). On the right-hand side of each plot, when this fraction is 1, we consider the entire test set (thus, the easy and hard subpopulations both have accuracy equal to the overall test accuracy).

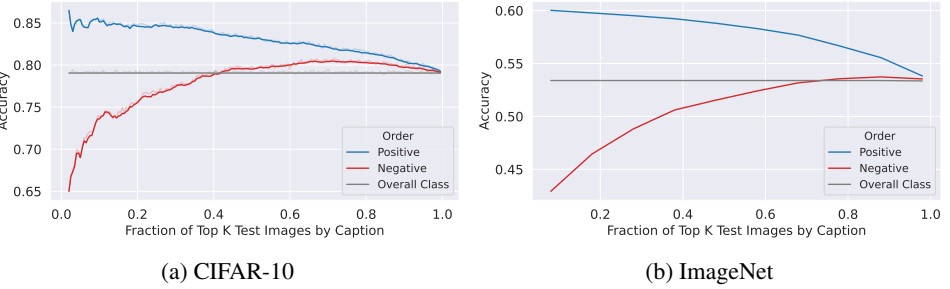

(a) CIFAR-10

(b) ImageNet

Figure 32: For CIFAR-10 and ImageNet, we plot the accuracy of the $K$ closest images to the positive or the negative caption for each class averaged over classes. On the x-axis, we plot the fraction of the test considered.

### C.2.3 DO FAILURE DIRECTIONS GENERALIZE TO OTHER ARCHITECTURES?

In this section, we perform our method on a second independent run of a ResNet18, as well as on a ResNet50. In Figure 33, we then plot the accuracies of the ImageNet test images that are closest to the extracted positive and negative captions. We find that our method behaves similarly on these models as in Figure 10b.

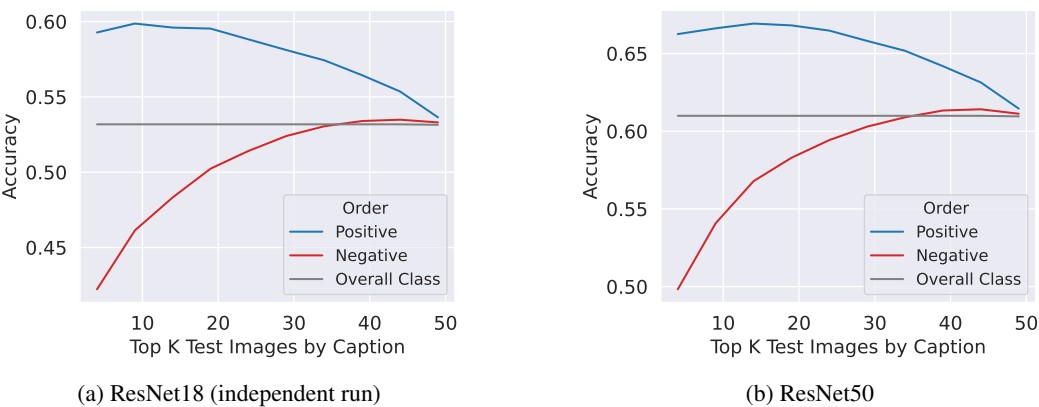

(a) ResNet18 (independent run)         (b) ResNet50

Figure 33: We run our method on (**a**) an independent run of a ResNet18 and (**b**) a ResNet50. As in Figure 10b, we plot the accuracy of the top K images closest to the positive and negative captions. The identified hard subpopulation has a lower accuracy than the identified easy subpopulation.

In Figure 34, we find that on a few example classes, the failure modes extracted by our method for a ResNet50 are similar to those extracted by the ResNet18 (see Figures 10a and 31).

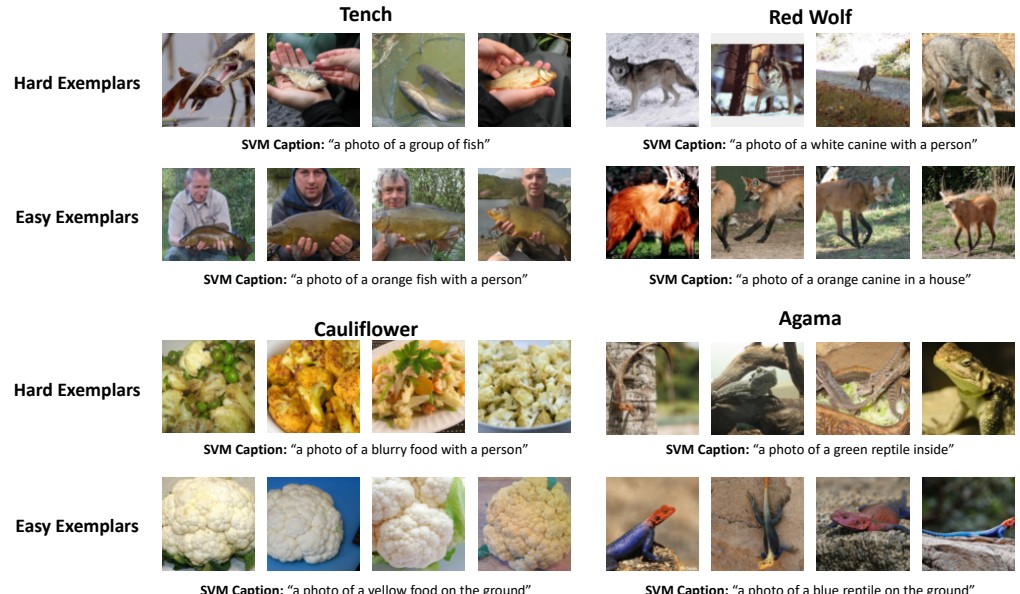

Figure 34: We display the most extreme examples and SVM captions when running our method for a ResNet50. We find that the failure directions are very similar to those extracted for a ResNet18.

Do the failure directions that we extracted using the ResNet18 also capture failure modes for other architectures? To answer this question, we measure the subpopulation accuracies of each of these models (as well as a pre-trained ViT-B) when using the ResNet18 positive/negative SVM captions to define our subpopulations of interest (Figure 35). We find that these ResNet18 captions perform almost as well as the SVM captions extracted for the specific underlying model, and define clear

"easy" and "hard" subpopulations. Thus, the directions extracted for the ResNet18 also represent real failure modes for other architectures.

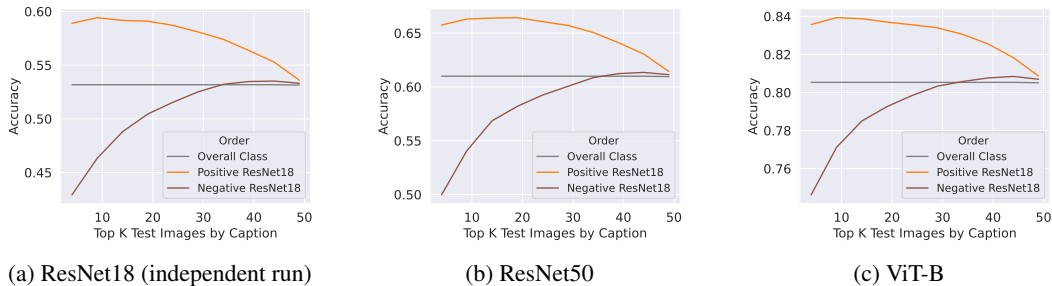

(a) ResNet18 (independent run)  (b) ResNet50  (c) ViT-B

Figure 35: We display the accuracies for various architectures of the test images closest to the SVM captions extracted from a *ResNet18*. The directions extracted for the ResNet18 represent real failure modes for other architectures.

### C.2.4  RESULTS ON FULL IMAGENET SPLIT

In Figure 36, we additionally run our method on an ImageNet split of 80% train and 20% validation (without any extra data). We find that our method behaves similarly to Figure 10b.

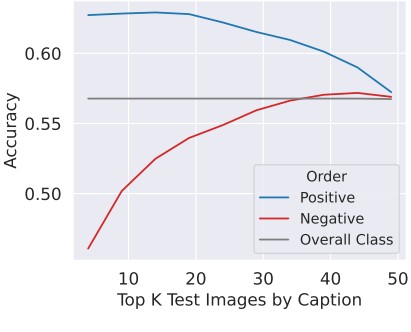

(a) ResNet18 (independent run)

Figure 36: We run our model on an 80% train, 20% validation split of ImageNet, and display the accuracies of the test images closest to the positive and negative caption.

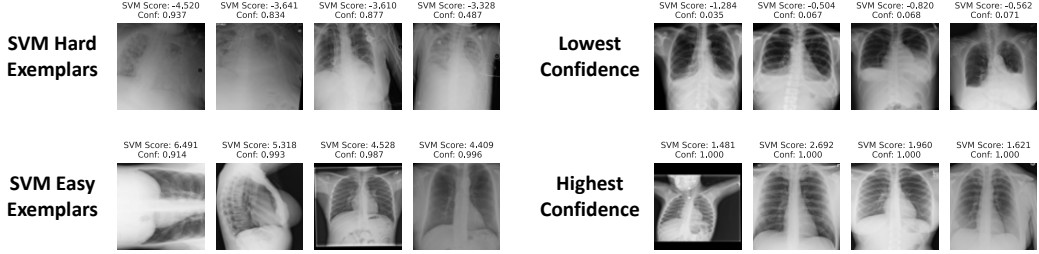

Figure 37: The most extreme examples by SVM decision value and by confidence for the "no effusion" class. Notably, the SVM distills a failure mode that is not easily seen in the low confidence examples.

| | | |
|---|---|---|
| *% Overall that is* | AP | 38.18% |
| | Incorrect | 12.62% |
| | Flagged | 28.11% |
| *% Incorrect that is AP* | | 55.351 % |
| *% Flagged that is AP* | | 59.826 % |

Figure 38: The fraction of examples from the "no effusion" class that are taken in the AP position. Notably, the base model disproportionately struggles on AP images, and the majority of images flagged by our framework are AP.

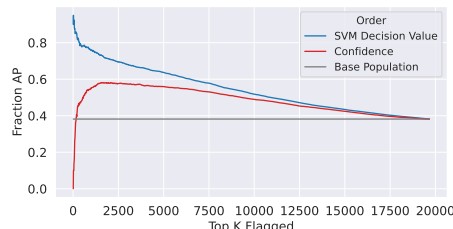

Figure 39: The fraction of the top K examples from "no effusion" class that are AP. We compare using the SVM's decision value and the model's confidences to select these images.

## C.3 CHESTX-RAY14

We apply our framework to the ChestXray-14 (Rajpurkar et al., 2017) dataset. This dataset contains frontal X-ray images labeled with 14 different conditions. ChestXray-14 is a multi-label tagging task (i.e., a single image can have more than one condition), so we treat each condition as its own binary problem. In this section, we focus on the condition Effusion. Results on other conditions can be found in Appendix C.3.

The trained SVM identifies visually distinguishable failure mode directions in latent space. As shown in Figure 37, the representative images flagged by this SVM as most incorrect are blurrier and less bright. Moreover, this trend is not reflected by the least confident images, indicating that our framework is isolating a different trend than the one corresponding to ordering the images by base model confidence.

In fact, we find that the SVM may be picking up on the *position* in which the exam was conducted. While the majority of the X-rays are Posterior-Anterior (PA) radiographs, a little over a third are Anterior-Posterior (AP). PA radiographs are usually clearer, but require the patient to be well enough to stand (Tafti & Byerly, 2020). Examples of AP and PA radiographs from the dataset can be found later in this section.

As shown in Table 38, the SVM for the class "no effusion" flags a large number of the AP radiographs as incorrect. This indicates that the model might indeed rely on the position in which the radiograph was taken to predict whether the patient was healthy. Moreover, the SVM selects the AP examples more consistently than ordering the radiographs by the base model's confidence (Figure 39).

Finally, in the rest of this appendix, we discuss the experimental details and additional results for applying our framework to the ChestX-ray14 dataset.

### C.3.1 EXPERIMENTAL DETAILS

We use the given train, validation, and test splits for the ChestX-ray14 dataset. We use a resolution of $224 \times 224$, and consider each of the 14 conditions separately.

**Hyperparameters**    We use the following hyperparameters, which are the same as for ImageNet.

| Parameter | Value |
|---|---|
| Batch Size | 1024 |
| Epochs | 16 |
| Peak LR | 0.5 |
| Momentum | 0.9 |
| Weight Decay | $5 \times 10^{-4}$ |
| Peak Epoch | 2 |

### C.3.2    EXAMPLES OF AP AND PA IMAGES

The ChestX-ray14 Images have different *positions* in which the exam could have been conducted. While the majority of the X-rays are Posterior-Anterior (PA) radiographs, a little over a third are Anterior-Posterior (AP).

PA radiographs are usually clearer, and the position of the scapula obstructs less of the lung; however, they require the patient to be well enough to stand (Tafti & Byerly, 2020). Examples of AP and PA radiographs from the dataset for the same patient and same resolution can be found in Figure 40. There are often also visible markers on the radiograph (i.e the words AP or portable) which distinguish the two.

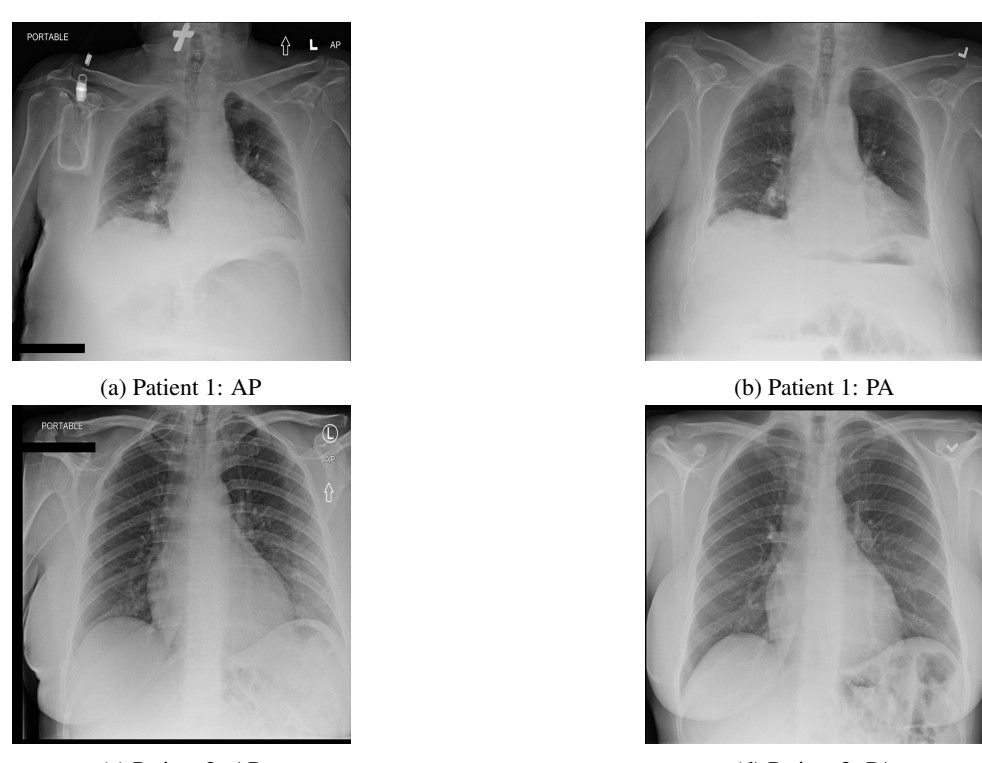

(a) Patient 1: AP                    (b) Patient 1: PA

(c) Patient 2: AP                    (d) Patient 2: PA

Figure 40: Examples of AP vs PA radiographs for the same patient.

### C.3.3 RESULTS FOR ADDITIONAL CONDITIONS

**Effusion** We begin with the condition "effusion". Figure 41 displays the most extreme figures by SVM decision value and confidence for each class. We additionally plot, for the top K values ordered by most negative decision value or confidence, the fraction of images that were AP radiographs. We find that, to a higher degree than confidences, the model associates AP images as hard for the healthy class and easy for the non-healthy class.

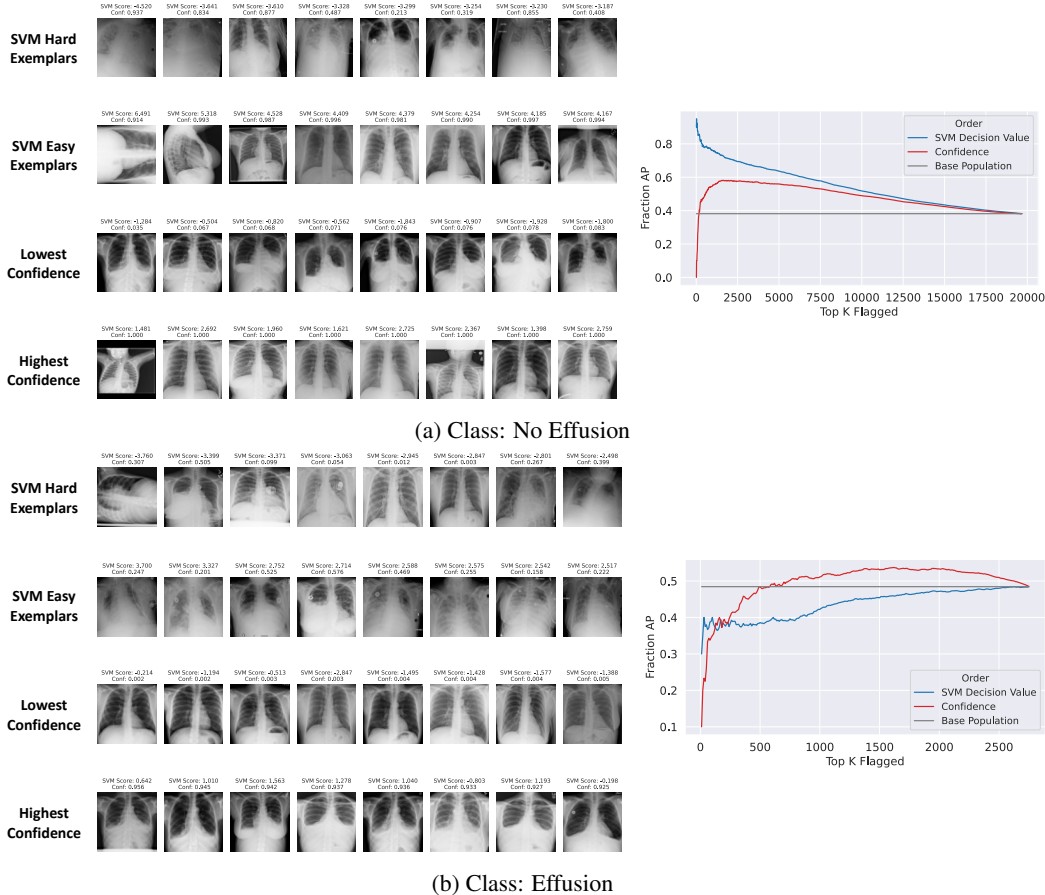

(a) Class: No Effusion

(b) Class: Effusion

Figure 41: **Effusion.** Left: Most extreme images according to SVM decision value and confidences. Right: The fraction of the top K images that were AP. We order images by the most negative SVM decision value or by the lowest model confidence.

**Mass**    We now consider the condition Mass (Figure 42). We again find that the SVM picks up on distinct visual patterns that were not surfaced by confidences. Moreover, we interestingly find that our framework surfaces AP images as hard for the *non-healthy* class, as opposed to the healthy class in the case of Effusion.

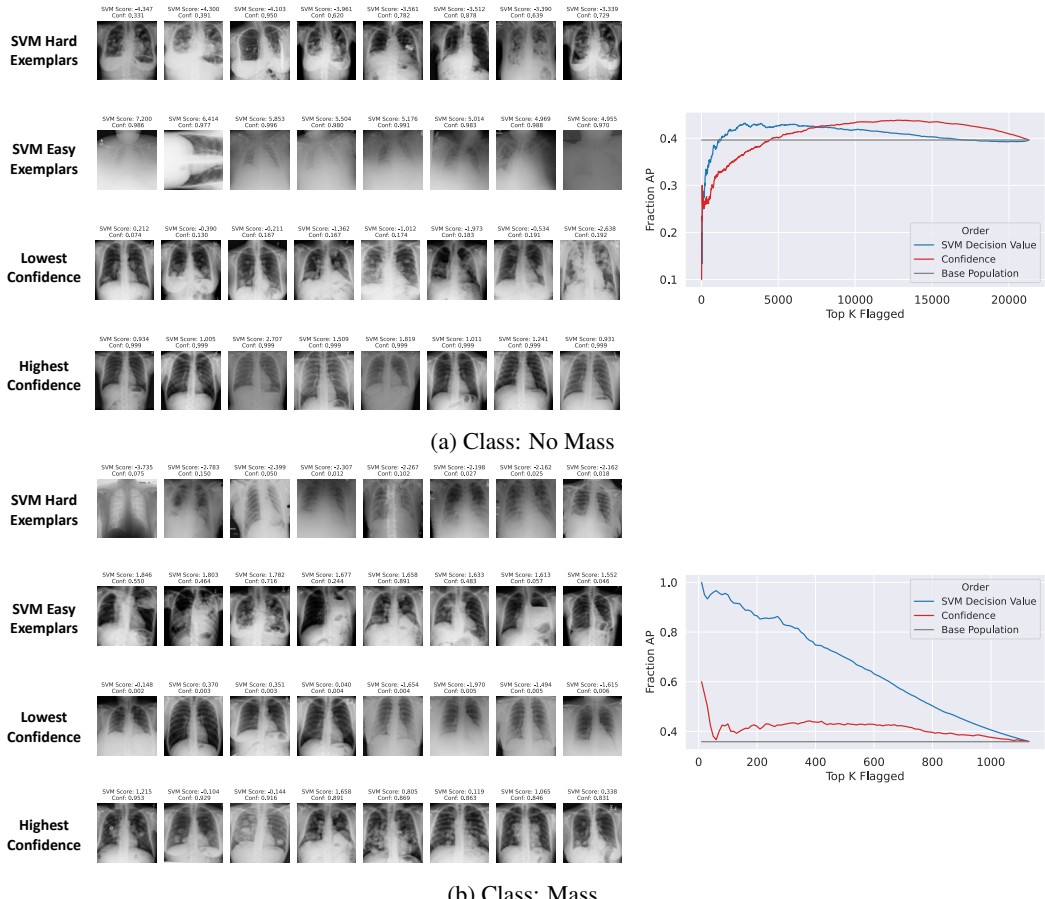

(a) Class: No Mass

(b) Class: Mass

Figure 42: **Mass.** Left: Most extreme images according to SVM decision value and confidences. Right: The fraction of the top K images that were AP. We order images by the most negative SVM decision value or by the lowest model confidence.

**Infiltration** Finally, we consider the condition Infiltration (Figure 43). As in the case of Effusion, the framework associates AP with easier images for the non-healthy class and, to a lesser extent, hard images for the healthy class.

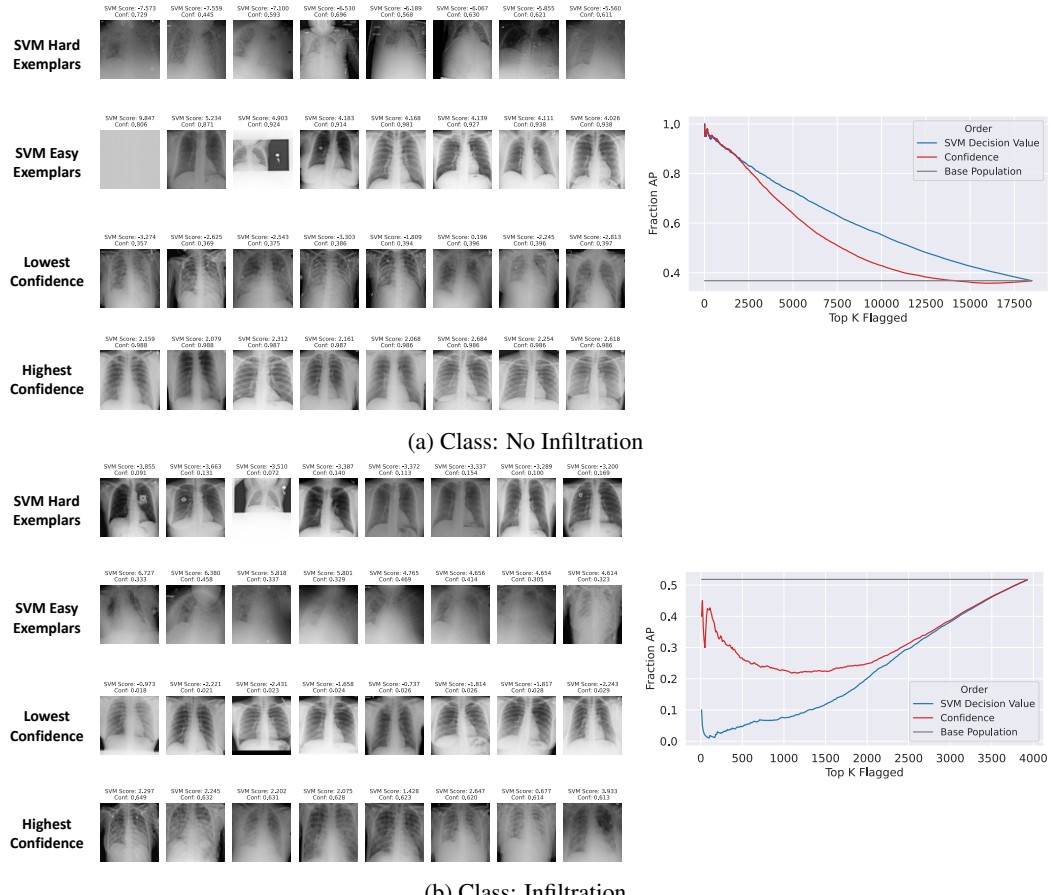

(a) Class: No Infiltration

(b) Class: Infiltration

Figure 43: **Infiltration.** Left: Most extreme images according to SVM decision value and confidences. Right: The fraction of the top K images that were AP. We order images by the most negative SVM decision value or by the lowest model confidence.

