# OpenReview forum: "Distilling Model Failures as Directions in Latent Space"
_ICLR.cc/2023/Conference — ICLR 2023 notable top 25%_

### Official Review · Reviewer_ySph · 2022-10-25

**Confidence:** 3
**Correctness:** 3
**Technical Novelty And Significance:** 2
**Empirical Novelty And Significance:** 3
**Recommendation:** 6

**Clarity, Quality, Novelty And Reproducibility:**

The writing is unclear as the motivation is unknown and the terms are undefined.
Training a SVM to predict whether the original model will classify a datapoint correctly has some limited novelty.


**Strength And Weaknesses:**

Strengths:
The paper proposes an automatic method to look for hard subpopulations and spurious correlations in datasets.
They performed extensive experiments to show their method identified the hard subpopulations and spurious correlations.

Weaknesses:
The captions and the images in result figures do not seem to match well:
In figure 1 easy exemplars, the man in the third image has a beard but the caption is "a photo of a old man who has no beard"
In figure 7 dog easy exemplars, the caption is "a photo of a white dog on the grass” but the images do not match
If the CLIP model is pretrained, based on the observation above, linear SVM seems not to be strong enough to separate the “easy” and “hard” examples for the original model
The writing of this paper prevents readers from understanding the paper:
The motivation is unclear, whether their ultimate goal is to look for error patterns or to improve the original model is unknown
The terms appearing in the abstract, such as “isolating hard subpopulations”, “spurious correlations ”, “distilling a model’s failure modes ”, ”directions within the feature space” …should be clearly defined early on.
In the main paper how to generate the captions is not mentioned. Although it appeared in the supplement but at least a brief description should be included in the main paper.


**Summary Of The Paper:**

The paper proposes an automatic method to look for hard subpopulations and spurious correlations in datasets,  by training SVMs in CLIP space to predict whether the original model will predict the data point correctly. Experiments show their method identified hard examples.


**Summary Of The Review:**

The writing prevents readers from understanding the paper and the novelty is somewhat limited.

----------------------------------------------
Post rebuttal comment:

After reading the other reviews and the rebuttal. We raise the score to 6.

---

> ### Author Response · Authors · 2022-11-09
> **Response to Reviewer ySph**
>
> We thank the reviewer for their response, and address their questions below.
>
> **[Clarification of the method and motivation]**
>
> In this work, our main contribution is to identify coherent subpopulations on which a given trained model has low accuracy (“hard subpopulations”). Such subpopulations may arise as a result of a spurious correlation, i.e. a decision rule which is predictive on the training set, but not in the real world or on the test set. Alternatively, they may simply be examples which are particularly complicated, underrepresented, or otherwise difficult for the trained model.
>
> We identify such subpopulations as directions in CLIP space, by training linear SVMs to distinguish between correct and incorrect instances for each class. We do not expect the SVMs to  succeed perfectly on this task, as the correct and incorrect instances are generally not linearly separable. In fact, this is the very reason for using a simple classifier -- so that it cannot overfit, and instead can only identify simple, interpretable attributes, rather than capturing every error a model makes. As we find, the SVM is able to detect minority populations more consistently than other methods (Figure 2), and is powerful enough to discover challenging subpopulations in large datasets such as ImageNet and ChestXray. We can then use the extracted directions to improve model reliability on these subpopulations (Figures 6 and 9).
>
> **[Captions in Figures 1 and 7]**
>
> Once we have trained our SVMs to predict errors, the vector normal to the SVM hyperplane is the direction of failure; this vector is the primary output of our method. The SVM captions are simply those whose text embeddings are closest to this extracted direction. The most extreme examples are the individual test images whose embeddings were most aligned to the failure direction.
>
> If the caption contains multiple failure modes, the most extreme example might not touch on all of them. For example, in Figure 7a, the easy caption is "a photo of a white dog on the grass." When we generate "easy" synthetic images directly from the failure direction (skipping the captioning step entirely) using a diffusion model, we do indeed generate white dogs on grass (Figure 8a): thus, the caption correctly describes the overall failure direction output by the SVM. However, by visual inspection, the easiest examples (according to the SVM) are indeed white dogs, but not necessarily on grass. Here, since these images fulfill one failure mode (color) particularly well, they are very aligned with the extracted direction, even though they do not fulfill the other failure mode (background).
>
> We check that our captions correctly capture planted correlations (e.g., "old man" and "young woman" in Figure 1) and define subpopulations that are confirmed to be challenging for the original model (Figure 7b, 10b). Perhaps most importantly, the captions are powerful enough to enable effective synthetic data augmentation, as shown in Figure 9.
>
> Natural language does have limits: some distribution shifts, such as pose, are very hard for even a human to describe in sentences. This was our main motivation for directly decoding the SVM direction using a diffusion model (bypassing the captioning step) to directly generate hard examples (see Figure 8).

---

### Official Review · Reviewer_ZiBp · 2022-10-26

**Confidence:** 4
**Correctness:** 4
**Technical Novelty And Significance:** 4
**Empirical Novelty And Significance:** 4
**Recommendation:** 8

**Clarity, Quality, Novelty And Reproducibility:**

This paper is well-written in detail and easy to understand. Moreover, the proposed method has clear novelty and originality.

**Strength And Weaknesses:**

Strength
1. The proposed method is novel. It is highly impressive 1) to utilize CLIP embeddings to make the proposed method interpretable and 2) to demonstrate the effect of data augmentation for the downstream tasks using pre-trained generative models based on the results of the proposed method.

2. The paper is well-written and easy to follow. Also, the appendix and figures help to understand the proposed method and experimental results easily. Especially, Figures 2 and 7-(b) enhance the reliability of the proposed method.

Weakness
1. My main concern is about the scalability of the proposed method. I think the experimental settings of Section 3 and Section 4.1 seem to be limited. Moreover, based on the results for ImageNet in Section 4.2 and Figure 27 in the appendix, it seems that the proposed method does not perform well with this dataset. Specifically, I think that the results of Figure 10-(a) and Figure 27 do not match well with the SVM caption. Also, in Figure 10-(b), the gap between the positive and negative samples decreases too quickly, unlike Figure 7-(b). Could the authors provide more evident results for the claim that the proposed methods are scalable?

2. In the Detection part of Section 2.1, the authors claim that the cross-validation score can be used as a measure of the failure model's strength in datasets. However, it seems that there is no experiment to validate its practicality. Could the authors verify the practicality of this score rather than the result of the correlation in Figure 4?


**Summary Of The Paper:**

This paper proposes a simple, yet effective method for identifying hard subpopulations in datasets. In addition, by utilizing the pre-trained CLIP and generative models, the underlying error patterns for identifying hard subpopulations become human-understandable. Based on such interpretations, appropriate data augmentations are applied to hard subpopulations, enhancing the performance of downstream tasks. Finally, the effects of the proposed method are validated for both cases where spurious correlation and the underrepresentation problem exist in the datasets.

**Summary Of The Review:**

The proposed method is novel and interesting. In addition, the experimental results are impressive overall. I hope that the authors will address two concerns during the rebuttal period.

---

> ### Author Response · Authors · 2022-11-09
> **Response to Reviewer ZiBp**
>
> We thank the reviewer for their response, and address their questions below.
>
> ### Questions on ImageNet Results:
> **[Figure 7b vs Figure 10b]**
>
> The ImageNet results of Figure 10b actually do match the CIFAR-10 results of Figure 7b. The reason that the gap appears to decrease more quickly in the Imagenet setting is due to the difference in the size of the test set.
>
> Specifically, we use the “val split” of ImageNet for the test set, which has only 50 examples per class (50,000 examples total). That means the right end of the x-axis of Figure 10b corresponds to considering all test examples of that class. Thus, if we consider “the top 50 examples closest to the positive or negative caption”, we are actually considering all of the test examples of that class (since there were only 50 examples to begin with). In that case, both the easy and hard subpopulations will have the accuracy of the base population (the intersection point of the two curves, at the rightmost point of the plot). In contrast, the CIFAR-10 test set has 1000 images per class, so the end of the x-axis of Figure 7b corresponds to considering only 20% of the test images for each CIFAR-10 class. As a result, the decay of the blue “positive” curve appears to be slower, and the two curves do not intersect.
>
> Put alternatively, considering the top 20% of the test set closest to the positive/negative caption corresponds to 10 images for ImageNet and 200 images for CIFAR-10. In Appendix C.2.2, we’ve added equivalent plots that put the x-axis in terms of % of the test dataset (rather than raw numbers of images). In this normalized version, you can see that our method on ImageNet and CIFAR-10 have similar behavior.
>
> **[ImageNet Captions]**
>
> The SVM captions are those whose text embeddings are closest in alignment to the extracted failure direction. The most extreme examples are the top 3 examples of the test set that were most aligned to the failure direction. Usually, this means that the caption will describe the three images well, but sometimes that mapping is not perfect.
>
> If the caption contains multiple failure modes, the hardest example might not touch on all of these failure modes. For instance, suppose the extracted caption is "a photo of a white horse in a stable." Here, an image of an especially white horse might be the test example that aligns best with the failure direction (even if it is not in a stable).  Moreover, the candidate caption set might not even contain the perfect caption: in the example of cauliflower in Appendix C.2.1, the best caption might include “on a plate”, but that phrase isn’t in our caption set. Nonetheless, in Figure 10b, we show that the images closest to the negative captions in ImageNet are harder than the images closest to the easiest captions, and thus represent real failure modes in the dataset.
>
> Natural language does have limits: some distribution shifts, such as pose, are very hard for even a human to describe in sentences. Indeed, this was our main motivation for directly plugging in the SVM direction into a diffusion model (bypassing the captioning step) to directly generate hard examples (see Figure 8).
>
> ### Further results on Detection
>
> In the newly added Appendix B.4.1, we provide a deeper exploration of the downstream implications of the cross-validation (CV) score in the CIFAR-100 case (Figures 22-24). Specifically, we instantiate the CIFAR-100 setup by underrepresenting the minority subpopulation with varying degrees. We find that the CV score is a good measure of the degree of underrepresentation. Moreover, the CV score is also indicative of the original model’s performance on the minority subpopulation.
>
> Since the CV score is correlated with the strength of the failure mode, it also reflects how many of the errors predicted by the SVM are actually part of the planted subpopulation. We further investigate the CV scores of individual classes (at a fixed degree of underrepresentation), and find that classes with a higher CV score appear to indicate a greater shift, in that a larger portion of the errors are from the minority subpopulation.

---

> > ### Comment · Reviewer_ZiBp · 2022-11-12
> > **Response**
> >
> > I appreciate the authors' responses and revisions of the paper. My concerns are well-addressed, so I raise my score to '8: Accept'. If time allows, could the authors discuss whether the detected biases in a particular dataset (e.g., ImageNet) are consistent in all experiments using a particular architecture such as ResNet? Additionally, I wonder if the same biases illustrated in the paper are detected when using models other than ResNet (e.g., ViT).

---

> > > ### Author Response · Authors · 2022-11-18
> > > **Response to Reviewer ZiBp**
> > >
> > > Thank you for raising your score!
> > >
> > > We have just added a new Appendix C.2.3. We ran our method on a second independent run of a ResNet18 and a ResNet50 on ImageNet, and find that our method similarly captures hard and easy subpopulations.
> > >
> > > Furthermore, we find that the SVM captions extracted from our original ResNet18 model also define challenging subpopulations for these other architectures. These failure modes even generalized to a pre-trained ViT-B. Thus, the directions extracted for the ResNet18 also represent real failure modes for other architectures.
> > >
> > > Finally, we inspected a few ImageNet classes by hand, and found that the identified "hardest" and "easiest" images from the ResNet50 visually match the analogous results from the ResNet18. (For example, the easy instances of the "tench" class remain tenches held by people).

---

### Official Review · Reviewer_HQAk · 2022-10-26

**Confidence:** 4
**Correctness:** 3
**Technical Novelty And Significance:** 3
**Empirical Novelty And Significance:** 3
**Recommendation:** 8

**Clarity, Quality, Novelty And Reproducibility:**


I find the proposed method to be novel and effective.

The paper is clear and well-written.

**Strength And Weaknesses:**

The proposed method is novel and opens potential for new line of research in identification of biases and difficult subpopulations. I personally find the submission to be clear and well-written. Overall, I am positive and would like to give a "accept".
However, I have few more comments that I hope to help improve the paper. Several of them are around the modeling choices and some of them are to clarify my understanding of the practical usefulness of the proposed method.
1.	The framework uses a linear support vector machine (SVM) to separate correct from incorrect examples and claims that the direction of the failure mode will be orthogonal to this decision boundary. My concern is whether the correct and incorrect examples are linearly separable. Can the authors provide data to support this claim? If it is not completely linearly separable, I suggest that the framework can be extended to use the simplest nonlinear classifiers.
2.	I find that the SVM will reduce the accuracy of the major subclasses (Table 1 in the Supplementary material). I suggest that the author discuss this phenomenon in the main text.
3.	I am concerned that the effectiveness of the proposed method depends on the data distribution of the validation set. My first request is for the authors to provide more experiments to verify the influence of long-tailed classes on the decision boundary of SVM. Second, I would like the authors to report the results with different proportions of dataset partitioning (e.g. 80% for training and 20% for validation).


**Summary Of The Paper:**

Deep learning models often exhibit consistent error patterns, where these errors often correspond to hard subpopulations in the data they are deployed on. This paper introduces a framework for automatic distillation and surfacing of a model’s error patterns. In detail, this framework uses SVM to predict the error of the original model in the CLIP’s embedding space, where the vector orthogonal to the decision boundary of SVM represents the direction of the failure mode of the original model. In addition, the authors also propose actionable interventions to remedy the failure mode.

The evaluation is done on hard subpopulations in image datasets such as CIFAR-10, ImageNet, and ChestX-ray14. Notably, the proposed method does not require direct
human intervention or pre-annotated subgroups and is therefore a scalable approach.


**Summary Of The Review:**

I have a few comments around modeling designs but I think the comments are addressable.

---

> ### Author Response · Authors · 2022-11-09
> **Response to Reviewer HQAk**
>
> We thank the reviewer for their feedback, and respond to each point below.
>
> **[1: Linear separability of the examples]**
>
> The data to the SVM classifier is not, in general, linearly separable -- in fact, it is not linearly separable except perhaps in the simplest datasets. We use a soft-margin SVM, which penalizes mis-classifications while maximizing the margin. Indeed, we can quantify the strength of the failure mode by the SVM’s ability to accurately (but still imperfectly) separate the mistakes from the correct examples. Figure 4 illustrates that the SVM’s validation accuracy is highly correlated with the strength of the spurious correlation (but still tends to be less than 100%). Also, if there is no consistent pattern of errors, naturally the SVM will not be able to accurately distinguish mistakes from correct examples.
>
> A key feature of our method is that, by using a simple classifier, we avoid overfitting to the error pattern, and thus capture simple and coherent error trends. The reviewer is correct that we could perhaps hope to use a very simple nonlinear classifier instead of an SVM. However, we find that an SVM works well in practice, and has the added benefit of imparting a useful geometry in the latent space. The SVM’s decision boundary is accessible in closed form; we can thus easily compute quantities like “distance from the decision boundary” to quantify the predicted difficulty of different examples.
>
> **[2: Accuracy reduction of the major subclasses]**
>
> As the reviewer points out, improving the accuracy of the hard subpopulations may reduce accuracy on majority groups. This phenomenon is typical even in interventions with perfectly annotated subpopulations, where the hard subpopulation is already known (see for example, [group DRO](https://arxiv.org/abs/1911.08731) or the oracle approach from Table 1). Indeed, this performance drop on the majority population is inevitable in the extreme case of a complete reliance on a spurious correlation (where the majority population might be perfectly predicted if using the spurious association, while the minority population is always incorrect).
>
> Thus, the goal of our interventions is to improve model reliability on challenging subpopulations without severely degrading performance on other populations. Indeed, all of our interventions either improve or maintain the original accuracy up to 1.5% point. In the revision, we will add a clarification on this important point to the main text.
>
> **[3: Data distribution of the validation set]**
>
> As the reviewer points out, the held-out validation set is an important component of our method. The fewer instances of the hard subpopulation in the validation set, the harder it is for the SVM to pick up on a coherent error pattern.
>
> However, we find that our method is relatively robust to the choice of validation set. In Appendix B.1.4, we apply our method with a validation set that matches the training distribution (and is thus heavily skewed). Even in this case, our method is able to successfully isolate the spurious correlation of gender. Moreover, in all of the natural datasets we evaluate (ImageNet, CIFAR-10, ChestX-ray14), the validation matches the training distribution.
>
> > My first request is for the authors to provide more experiments to verify the influence of long-tailed classes on the decision boundary of SVM.
>
> We are not sure what the reviewer means by “verify the influence of long-tailed classes on the decision boundary of the SVM”. Does Appendix B.1.4 answer this request, and if not, would it be possible to clarify what the requested experiment is?
>
> > Second, I would like the authors to report the results with different proportions of dataset partitioning (e.g. 80% for training and 20% for validation).
>
> As the reviewer requested, we have added further experiments on CIFAR-100 with 80% train, 20% test to Appendix B.4.1. We find that the results are consistent with the original Figure 5.

---

### Official Review · Reviewer_ZdLg · 2022-10-26

**Confidence:** 3
**Correctness:** 4
**Technical Novelty And Significance:** 3
**Empirical Novelty And Significance:** 3
**Recommendation:** 8

**Clarity, Quality, Novelty And Reproducibility:**

Paper was well written, in particular the appendix provides the necessary detail to recreate the results. Excellent use of the new generation of text2image style models to help improve model performance.

**Strength And Weaknesses:**

Strengths.

- The use of the SVM to create the boundary between the model's correct and incorrect outputs and having an image/caption style model's feature space provides a lot of capabilities to this approach. Captions can be provided to understand what the easiest versus hardest examples are, and the model allows hard and easy examples to be generated.
- Excellent observation that the SVM does better than using the model confidences to determine hard/easy exemplars.


Weakness
- I worried that the paper was too CLIP limited; would the approach have worked as well with a different feature space representation.
- In "Targeted synthetic data augmentation", the authors only retrained the last layer of the model but gave no reason as to why.

**Summary Of The Paper:**

The authors propose an automated way to determine a model's error modes (hard subpopulations and spurious correlations). They do so by using linear classifiers (SVM) on a feature space (CLIP) such that these failure modes are a direction within the feature space (orthogonal to the hyperplane created by the SVM). Given the feature space used, diffusion models can allow for the generation of additional samples to address the failure modes of the model. The authors use the CelebA dataset (which has a spurious correlation between gender and age) and the CIFAR-10 dataset (which has underrepresented subtypes).

**Summary Of The Review:**

A thorough a well written paper showing how SVM and CLIP feature space can be used to automatically identify hard subpopulations and spurious correlations. Would like the two weaknesses to be clarified. The technique proposed could provide an additional tool for model understanding and post-hoc improvement.

---

> ### Author Response · Authors · 2022-11-09
> **Response to Reviewer ZdLg**
>
> We thank the reviewer for their response, and address their two questions below.
>
> **[Different feature space representation]**
>
> As the reviewer points out, the quality of the latent space plays an important role in our method. For example, if the embedding does not capture the feature associated with a spurious correlation, our method is unlikely to detect such a bias. However, we found that most reasonable latent spaces work. In particular, in Appendix B.1.6, we explored the use of different feature space representations, including the model’s original latent space and Inceptionv3 features. Employing our method with Inception embeddings leads to only slightly worse results than employing CLIP, so our approach is relatively robust to different feature space representations.
>
> We find overall that different featurizations are effective in our approach, so long as they are agnostic to the particular training task at hand (e.g. not the model’s own latent space). Also, it is worth noting that using CLIP has the added advantage of embedding language in the same latent space (and thus allowing us to automatically caption the extracted directions in latent space and interface with text-to-image diffusion models).
>
> **[Retraining the last layer of the model]**
>
> We found that, in practice, fine-tuning the last layer of the model was most effective at improving performance on the hard-subpopulation. This is in agreement with [recent work](https://arxiv.org/abs/2202.10054), which found that fine-tuning the last layer is more effective than fine-tuning the full model in cases of distribution shift.

---

### Author Response · Authors · 2022-11-09
**Updates in Revision**

We thank all reviewers for their thoughtful feedback. We have made a few additions to the revision in response, and summarize them here:

1) We have added Appendix B.4.1, which explores a CIFAR-100 setting with an 80% train and 20% test split. Furthermore, in this setting, we provide deeper analysis of the downstream implications of the CV score, demonstrating that it is highly correlated with the fraction of errors predicted by the SVM that are part of the (known) hard subpopulation.
2) In Appendix C.2.2, we plot the model’s accuracy on the fraction of the test set closest to the positive or negative caption for ImageNet and CIFAR-10. These are equivalent to Figures 7b and 10b, except normalized for test set size.

We hope that these additions, in addition to the individual responses, address all concerns raised by the reviewers.

---

### Decision · Program_Chairs · 2023-01-20

**Decision:**

Accept: notable-top-25%

**Justification For Why Not Higher Score:**

The scope of this work is about model diagnosis and intervention, it is important but the audience is a bit small.

**Justification For Why Not Lower Score:**

Overall this is a well-executed paper.  The reviewers liked the idea and the paper showed that the idea works.

**Metareview: Summary, Strengths And Weaknesses:**

This paper proposed an approach to identify, interpret and act on a learned model’s failure modes.  The idea is to use a simple linear classifier (a linear SVM) in the feature space to classify examples that the model can predict well vs not.  This linear classifier is then used to identify the most typical examples of the failure mode, interpret the failure mode by e.g. identifying the most typical examples or synthesizing typical examples from the feature space, which can then be used to improve training, by adding hard examples or synthetic examples to the training set.

It is good to see works like this to improve machine learning model’s robustness.  All reviewers favored acceptance after the rebuttal, and the authors have also improved the draft and addressed reviewers’ concerns.  I support acceptance.

**Note From Pc:**

if the above contains the word "oral" or "spotlight" please see: "oral" presentation means -> notable-top-5% and "spotlight" means -> notable-top-25%. As stated in our emails, we are disassociating presentation type from AC recommendations